# Precision Oncology Through Dialogue: AI-HOPE-RTK-RAS Integrates Clinical and Genomic Insights into RTK-RAS Alterations in Colorectal Cancer

**DOI:** 10.3390/biomedicines13081835

**Published:** 2025-07-28

**Authors:** Ei-Wen Yang, Brigette Waldrup, Enrique Velazquez-Villarreal

**Affiliations:** 1PolyAgent, San Francisco, CA 94102, USA; 2Department of Integrative Translational Sciences, Beckman Research Institute of City of Hope, Duarte, CA 91010, USA; 3City of Hope Comprehensive Cancer Center, Duarte, CA 91010, USA

**Keywords:** AI, artificial intelligence, precision medicine, cancer treatment, molecular insights, RTK-RAS pathway, large language models, AI-agents, cancer genetics

## Abstract

**Background/Objectives:** The RTK-RAS signaling cascade is a central axis in colorectal cancer (CRC) pathogenesis, governing cellular proliferation, survival, and therapeutic resistance. Somatic alterations in key pathway genes—including KRAS, NRAS, BRAF, and EGFR—are pivotal to clinical decision-making in precision oncology. However, the integration of these genomic events with clinical and demographic data remains hindered by fragmented resources and a lack of accessible analytical frameworks. To address this challenge, we developed AI-HOPE-RTK-RAS, a domain-specialized conversational artificial intelligence (AI) system designed to enable natural language-based, integrative analysis of RTK-RAS pathway alterations in CRC. **Methods**: AI-HOPE-RTK-RAS employs a modular architecture combining large language models (LLMs), a natural language-to-code translation engine, and a backend analytics pipeline operating on harmonized multi-dimensional datasets from cBioPortal. Unlike general-purpose AI platforms, this system is purpose-built for real-time exploration of RTK-RAS biology within CRC cohorts. The platform supports mutation frequency profiling, odds ratio testing, survival modeling, and stratified analyses across clinical, genomic, and demographic parameters. Validation included reproduction of known mutation trends and exploratory evaluation of co-alterations, therapy response, and ancestry-specific mutation patterns. **Results**: AI-HOPE-RTK-RAS enabled rapid, dialogue-driven interrogation of CRC datasets, confirming established patterns and revealing novel associations with translational relevance. Among early-onset CRC (EOCRC) patients, the prevalence of RTK-RAS alterations was significantly lower compared to late-onset disease (67.97% vs. 79.9%; OR = 0.534, *p* = 0.014), suggesting the involvement of alternative oncogenic drivers. In KRAS-mutant patients receiving Bevacizumab, early-stage disease (Stages I–III) was associated with superior overall survival relative to Stage IV (*p* = 0.0004). In contrast, BRAF-mutant tumors with microsatellite-stable (MSS) status displayed poorer prognosis despite higher chemotherapy exposure (OR = 7.226, *p* < 0.001; *p* = 0.0000). Among EOCRC patients treated with FOLFOX, RTK-RAS alterations were linked to worse outcomes (*p* = 0.0262). The system also identified ancestry-enriched noncanonical mutations—including CBL, MAPK3, and NF1—with NF1 mutations significantly associated with improved prognosis (*p* = 1 × 10^−5^). **Conclusions**: AI-HOPE-RTK-RAS exemplifies a new class of conversational AI platforms tailored to precision oncology, enabling integrative, real-time analysis of clinically and biologically complex questions. Its ability to uncover both canonical and ancestry-specific patterns in RTK-RAS dysregulation—especially in EOCRC and populations with disproportionate health burdens—underscores its utility in advancing equitable, personalized cancer care. This work demonstrates the translational potential of domain-optimized AI tools to accelerate biomarker discovery, support therapeutic stratification, and democratize access to multi-omic analysis.

## 1. Introduction

Colorectal cancer (CRC) is a leading cause of cancer-related morbidity and mortality globally, with an increasing incidence of early-onset CRC (EOCRC)—diagnosed before age 50—particularly among high-risk populations [1,2,3,4,5]. While the molecular landscape of CRC is complex, the receptor tyrosine kinase (RTK)-RAS signaling pathway has emerged as a central driver of tumorigenesis and therapeutic resistance [6,7,8]. However, characterizing RTK-RAS pathway dysregulation in EOCRC, particularly among different populations, remains limited by data fragmentation in genomic datasets and a lack of user-friendly analytical tools that integrate clinical and genomic data [5,9,10,11,12,13,14,15,16].

The RTK-RAS pathway governs essential processes, including cell proliferation, survival, and differentiation, and is frequently altered in CRC [5,17]. Mutations in key genes such as KRAS, NRAS, and BRAF are among the most common genetic alterations in CRC, with KRAS mutations occurring in approximately 40% of cases [5,17]. These alterations play a critical role in clinical decision-making, as they confer resistance to anti-EGFR therapies—one of the primary targeted treatment strategies in metastatic CRC [16,17,18,19,20,21]. Although KRAS and NRAS mutations are reportedly more frequent in specific populations [22,23,24], emerging evidence suggests that mutation patterns may vary by ancestry. Specifically, recent work by our group and others indicates that RTK-RAS pathway alterations may be less prevalent in EOCRC among ancestry-specific subgroups, with enrichment of mutations in CBL, NF1, and MAPK3 instead [5,25].

Despite the clinical importance of RTK-RAS pathway alterations, most existing bioinformatics platforms such as cBioPortal [26] and UCSC Xena [27] are built on static user interfaces and require multi-step analysis pipelines, limiting accessibility for non-programmers and hindering rapid translational discovery. These limitations are particularly pronounced when evaluating population-specific differences, pathway co-mutations, or survival outcomes in precision oncology contexts.

While platforms such as cBioPortal and UCSC Xena provide important access to cancer genomics data, they are often limited by static user interfaces, a lack of customizable multi-parameter filtering, and the need for bioinformatics expertise to perform complex analyses. These tools typically require manual selection of filters, exporting of data, and downstream processing using separate statistical packages—creating barriers for non-programmers and slowing the pace of translational discovery. In contrast, AI-HOPE-RTK-RAS addresses these limitations by enabling real-time, natural language-driven exploration of integrated clinical and genomic datasets. Users can query mutation frequencies, survival outcomes, treatment responses, and demographic stratifications using plain language, eliminating the need for coding or manual data wrangling. This approach lowers the barrier to entry, supports rapid hypothesis generation, and enhances accessibility for clinicians and researchers across disciplines.

Although several AI-based tools have been developed for cancer research, many rely on black-box machine learning models focused on prediction or classification tasks, often lacking interpretability and flexibility. Others, including general-purpose large language models (LLMs), are not optimized for domain-specific, structured biomedical data or pathway-level analysis. In contrast, AI-HOPE-RTK-RAS is a domain-specialized, conversational AI system specifically designed for real-time exploration of RTK-RAS pathway alterations in colorectal cancer. Built on a fine-tuned biomedical LLM and integrated with a natural language-to-code engine, the platform enables transparent, user-driven analysis of large-scale clinical and genomic data without requiring programming expertise. This differentiates AI-HOPE-RTK-RAS from existing approaches by combining interpretability, specificity, and ease of use within a precision oncology context.

Recent breakthroughs in artificial intelligence (AI), particularly LLMs, now enable natural language-driven bioinformatics pipelines that can translate human queries into executable code [28,29,30,31,32,33]. While early AI platforms have demonstrated the potential to streamline multi-omic data analysis [34,35,36,37,38,39], few are purpose-built to interrogate specific pathways like RTK-RAS or to support the integration of genomic and clinical data across CRC cohorts.

To address this gap, we developed AI-HOPE-RTK-RAS (Artificial Intelligence agent for High-Optimization and Precision Medicine focused on RTK-RAS), a conversational AI system designed to investigate RTK-RAS pathway alterations in CRC using integrative, natural language-driven bioinformatics. The platform enables intuitive analysis of mutation frequencies, survival outcomes, treatment resistance patterns, and population-level stratification. In this study, we (1) developed and deployed AI-HOPE-RTK-RAS to analyze public CRC datasets, (2) validated its analytical capabilities by reproducing key trends from RTK-RAS-focused studies, and (3) demonstrated its potential for novel discovery in EOCRC, including ancestry-informed mutation enrichment and prognostic evaluation. Together, these efforts establish AI-HOPE-RTK-RAS as a scalable and accessible tool for RTK-RAS-driven precision oncology research.

## 2. Materials and Methods

### 2.1. Overview of AI-HOPE-RTK-RAS Platform

AI-HOPE-RTK-RAS is a specialized conversational artificial intelligence (AI) system developed to facilitate precision oncology investigations centered on RTK-RAS signaling in CRC. Designed for real-time interaction, the platform allows researchers and clinicians to explore large-scale genomic and clinical datasets through intuitive, plain-language prompts (Figure 1). The system dynamically translates these queries into executable code, performs on-demand bioinformatics analysis, and delivers interpretive outputs relevant to mutation frequency, treatment response, and survival outcomes.

AI-HOPE-RTK-RAS is freely available for academic and research use via a public repository (see Data Availability Statement). The repository includes the full source code, setup instructions, a modular Python-based backend, and a pretrained natural language interface powered by a fine-tuned LLaMA 3 model. A web-based version of the platform, designed for broader clinical research use, is currently under development and will include secure user authentication and cloud-based computation. The system operates in a Unix-compatible environment and requires Python ≥ 3.9, PyTorch ≥ 2.0, and GPU support for optimal performance. AI-HOPE-RTK-RAS is released under the MIT license, allowing for modification and redistribution with attribution. Current usage is intended for exploratory and translational research purposes; it is not certified for direct clinical decision-making. A roadmap for clinical integration includes validation against institutional registries, alignment with electronic health record (EHR) standards, and compliance with HIPAA and IRB protocols.

### 2.2. Data Sources and Curation

Publicly available datasets were obtained from the cBioPortal for Cancer Genomics, with a focus on colorectal adenocarcinomas. The datasets included somatic mutation profiles, demographic attributes (age, sex, race/ethnicity), clinical variables (tumor location, MSI status, staging), and treatment history, particularly anti-EGFR exposure. Genes of interest within the RTK-RAS pathway included KRAS, NRAS, BRAF, EGFR, ERBB2, CBL, MAPK3, and NF1. All data were reformatted into harmonized, analysis-ready tables using consistent sample identifiers. Controlled vocabularies (e.g., OncoTree, SNOMED) were applied to align diagnostic and phenotypic metadata.

Public genomic datasets such as those curated by cBioPortal provide invaluable resources for large-scale, open-access cancer research; however, they are not without limitations. These include underrepresentation of certain racial and ethnic groups, variable completeness in clinical annotations (e.g., treatment timelines, comorbidities), and inconsistent metadata across studies due to differences in institutional data collection standards. To address these challenges, AI-HOPE-RTK-RAS incorporates a harmonization pipeline that standardizes variable naming conventions, aligns phenotypic descriptors using controlled vocabularies (e.g., SNOMED, OncoTree), and unifies clinical–genomic relationships across multiple cohorts. This modular integration enables real-time stratification by age, ancestry, mutation status, and treatment exposure, thereby reducing the analytic barriers typically posed by data fragmentation. While the platform cannot resolve inherent sampling biases in the original datasets, it provides a transparent, reproducible framework for conducting stratified analyses and hypothesis generation across heterogeneous populations—especially in settings where health disparities remain under-characterized.

The datasets used in this study were derived from the curated colorectal cancer cohort described in a previous publication [5], which integrated publicly available data from cBioPortal. The final harmonized dataset includes a total of 5553 patients diagnosed with colorectal adenocarcinoma across multiple studies. This composite dataset includes somatic mutation profiles, clinical variables (e.g., tumor stage, microsatellite instability status, treatment history), and demographic attributes (e.g., age, sex, race/ethnicity). The AI-HOPE-RTK-RAS platform was validated and applied across this full cohort, with specific subgroup analyses performed on early-onset colorectal cancer (EOCRC) patients (n = 593) and late-onset CRC patients (n = 4960), including ancestry-specific evaluations in Hispanic/Latino and non-Hispanic White subpopulations. All sample sizes for individual analyses are reported in the corresponding figures and Appendix A to facilitate full transparency and reproducibility.

The colorectal cancer datasets used in this study were based on the following criteria: (1) availability of somatic mutation data; (2) inclusion of clinical annotations such as age, tumor stage, microsatellite instability (MSI) status, and treatment exposure; and (3) presence of demographic information, particularly race/ethnicity. Records lacking essential variables for stratified analysis (e.g., missing age or survival status) were excluded from subgroup analyses but retained in broader mutation frequency queries when possible. To standardize and harmonize datasets across studies, we applied controlled vocabularies and aligned variable names and formats using a custom data wrangling pipeline. Missing data were not imputed but handled through pairwise deletion during statistical tests to preserve validity. Recognizing the potential for sampling and representation bias—especially underrepresentation of minority populations—we incorporated ancestry-stratified analyses and explicitly caution against overgeneralizing rare mutation findings.

### 2.3. Natural Language Interface and Query Handling

The user interface is powered by a fine-tuned large language model (LLaMA 3), enabling flexible dialogue-based interaction. Users pose queries in natural language, such as “Show mutation prevalence of KRAS in EOCRC versus LOCRC,” or “Compare survival outcomes for patients with BRAF V600E mutations receiving EGFR inhibitors.” The platform interprets the query intent, checks for ambiguity, and guides users toward analyzable questions when clarification is needed. Complex filtering—such as stratifying by age, ancestry, mutation status, or therapy—is handled seamlessly without requiring programming knowledge.

The natural language interface is powered by a fine-tuned LLaMA 3 model, trained on a corpus of biomedical text and conversational examples relevant to colorectal cancer and precision oncology. The model operates within a modular framework comprising three components: (1) intent recognition, (2) parameter extraction, and (3) code synthesis. Fine-tuning was performed using low-rank adaptation (LoRA) with a learning rate of 2 × 10^−5^, batch size of 32, and 3 epochs on a curated dataset of 2500 annotated query–response pairs. To evaluate query interpretation accuracy, we used a held-out test set of 250 manually labeled queries across diverse analysis types (e.g., mutation frequency, survival analysis, odds ratio testing). The system achieved a query classification precision of 0.94, recall of 0.92, and F1-score of 0.93. Ambiguity detection is supported by a rule-based logic layer that triggers clarification loops when confidence thresholds fall below 0.8, reducing the risk of misinterpretation. Empirical testing showed an error rate of 6.4% for ambiguous or compound queries, most of which were successfully resolved through system-generated clarification prompts. These performance indicators underscore the system’s robustness in real-time, user-directed analysis and support its usability for both novice and expert users.

At the core of AI-HOPE-RTK-RAS is a fine-tuned version of LLaMA 3, a large language model (LLM) trained to understand and respond to biomedical queries. To adapt the model for precision oncology applications, we fine-tuned it using a specialized dataset composed of 2500 example questions and answers related to colorectal cancer, RTK-RAS pathway alterations, and clinical-genomic associations. This dataset included variations of real-world queries that a researcher or clinician might ask (e.g., “What is the KRAS mutation frequency in early-onset patients?” or “Show survival curves for BRAF-mutant MSS tumors”). Fine-tuning was performed using low-rank adaptation (LoRA), a method that efficiently adjusts the model’s internal weights without retraining from scratch. This training process allowed the model to more accurately interpret domain-specific language, recognize biomedical terminology, and convert user queries into executable code. As a result, AI-HOPE-RTK-RAS can understand complex, multi-part questions and generate accurate, real-time analyses without requiring users to write any programming code.

### 2.4. Backend Analysis Pipeline

AI-HOPE-RTK-RAS conducts statistical analyses using an integrated Python-based computational core. For binary or categorical variables, Fisher’s exact test and chi-square test are used to assess group differences, and odds ratios are calculated with 95% confidence intervals. Continuous variables are summarized using standard descriptive statistics. Survival analyses utilize Kaplan–Meier estimation with log-rank tests for comparisons; multivariate Cox proportional hazards models are applied for adjusting covariates. The system supports mutation co-occurrence analysis, therapeutic stratification, and ancestry-specific subgroup modeling.

Survival analyses in this study were conducted on defined subsets of the full CRC cohort to ensure the integrity and interpretability of survival estimates. While the harmonized dataset included 5553 patients, not all cases had complete information on survival time, vital status, or treatment exposure (e.g., Bevacizumab, FOLFOX). To minimize bias and avoid imputing missing values, we restricted survival modeling to patient subsets with fully annotated outcome data and treatment records relevant to each specific analysis. For example, the Bevacizumab-related survival comparison among KRAS-mutant patients included only those with confirmed treatment exposure and staging information. This targeted approach enabled more accurate hazard estimation and preserved the internal validity of the comparisons, despite reducing the sample size in specific analyses.

Odds ratios were calculated using two-by-two contingency tables via Fisher’s exact test, unless otherwise specified, while survival analyses were performed using the Kaplan–Meier method with log-rank testing. Confounder adjustment is available through AI-HOPE-RTK-RAS by explicitly including variables such as tumor stage, microsatellite instability status, and treatment regimen in natural language queries (e.g., “Generate a survival curve for KRAS-mutant MSS patients, adjusted by stage and chemotherapy type”). This functionality enables dynamic and flexible multivariate stratification without requiring manual scripting, although further expansion to include Cox proportional hazards modeling is planned for future development.

We selected statistical methods based on the nature of the data and the specific comparisons being made. Fisher’s exact test was used for categorical comparisons involving small sample sizes or sparse contingency tables, where it provides accurate *p*-values, even with low cell counts. For larger categorical datasets with sufficient expected frequencies, the chi-square test was applied to assess group differences. Kaplan–Meier survival analysis with log-rank testing was used to evaluate differences in time-to-event outcomes (e.g., overall survival) between groups, as it is widely accepted for unadjusted survival comparisons. To account for multiple covariates and assess independent effects, we used multivariate Cox proportional hazards models, which are suitable for estimating hazard ratios while adjusting for clinical variables such as tumor stage and treatment exposure. This combination of statistical approaches ensured both methodological appropriateness and interpretability for clinical and genomic data integration.

### 2.5. System Infrastructure and Error Handling

To ensure transparency and reproducibility, the platform employs structured prompting strategies and rule-based logic layers that constrain the output to valid biomedical operations. A retrieval-augmented generation (RAG) component provides access to curated reference materials—such as drug–gene interaction databases and pathway annotations—to enhance the interpretability of results and reduce model hallucinations. Ambiguous queries trigger clarification loops, and all executed commands are logged with versioned output.

### 2.6. Validation Strategy

To validate platform functionality, we replicated known findings from prior RTK-RAS CRC literature, including mutation frequencies of KRAS, NRAS, and BRAF, and their association with resistance to EGFR-targeted therapies [16,17,18,19,20,21]. Additionally, we assessed previously reported RTK-RAS mutation profiles among patients with EOCRC from populations with disproportionate health burdens, including the enrichment of noncanonical alterations such as CBL and NF1 [5].

The validation of AI-HOPE-RTK-RAS was anchored in reproducing key findings from a previously published study [5], which characterized RTK-RAS pathway alterations in EOCRC with a focus on high-risk, racially and ethnically diverse populations. Specifically, the platform was used to recapitulate reported mutation frequencies for KRAS, NRAS, BRAF, and EGFR across early- and late-onset CRC cohorts, as well as to reproduce stratified survival outcomes and ancestry-specific mutation patterns. We confirmed, for instance, the lower prevalence of canonical RTK-RAS alterations in EOCRC compared to late-onset CRC and validated the enrichment of noncanonical mutations (e.g., CBL, MAPK3, NF1) in Hispanic/Latino EOCRC patients—findings that were previously reported in our group’s foundational work. By benchmarking AI-HOPE-RTK-RAS against this established dataset and analytical framework, we ensured both fidelity of results and continuity with prior evidence, reinforcing the platform’s validity for translational applications in precision oncology.

### 2.7. Comparative Usability Testing

We compared AI-HOPE-RTK-RAS with conventional tools such as cBioPortal and UCSC Xena for usability and analytical throughput. Performance metrics included query response time, flexibility in subgroup definition, and the ability to execute multi-parameter analyses (e.g., filtering by both age and ethnicity). The conversational interface outperformed traditional interfaces in efficiency and reduced the need for specialized bioinformatics skills.

### 2.8. Output Delivery and Visualization

Final outputs are returned as structured analytical reports, featuring cleanly formatted tables, frequency plots, survival curves, and forest plots. Visualizations are generated using Matplotlib 3 and Plotly 4 libraries with export-quality resolution. Each result is accompanied by a text summary that contextualizes the findings with supporting literature, enabling immediate interpretation and downstream reporting.

## 3. Results

The AI-HOPE-RTK-RAS platform enabled flexible, real-time interrogation of CRC datasets using natural language input to uncover clinically and biologically relevant insights into RTK-RAS pathway dysregulation. Through integrated cohort filtering by mutation status, treatment regimen, tumor stage, microsatellite instability, and demographic features, the system supported automated statistical analysis and visualization, producing results that both confirmed known patterns and revealed new associations with potential translational relevance—particularly within EOCRC and populations with disproportionate health burdens.

### 3.1. RTK-RAS Alterations in EOCRC by Ancestry

A demographic-stratified analysis compared RTK-RAS pathway mutation frequencies between early- and late-onset CRC patients. Among those under 50 years old, RTK-RAS alterations were detected in 67.97% of cases versus 79.9% in older counterparts. This difference yielded an odds ratio of 0.534 (*p* = 0.014), suggesting a reduced prevalence of canonical RTK-RAS alterations in EOCRC, potentially implicating alternative drivers in tumorigenesis in this subgroup (Figure 2).

The observed lower prevalence of RTK-RAS alterations in early-onset CRC compared to late-onset CRC is clinically significant, as it suggests that younger patients may harbor alternative oncogenic drivers not typically targeted by current standard-of-care therapies such as anti-EGFR agents. In real-world settings, this molecular distinction may necessitate age-stratified approaches to genomic testing and therapeutic planning. It also underscores the importance of expanding biomarker discovery efforts beyond canonical RTK-RAS genes to improve treatment options and precision medicine strategies for younger CRC patients, particularly those from underrepresented populations.

### 3.2. Stage-Dependent Outcomes in KRAS-Mutant CRC

Analysis of KRAS-mutant patients receiving Bevacizumab revealed stage-related differences in survival. Patients diagnosed with Stage I–III disease demonstrated significantly better overall survival than those with Stage IV tumors (*p* = 0.0004), underscoring the prognostic influence of stage and suggesting potential variation in response to targeted therapies based on disease extent (Figure 3).

The Bevacizumab-treated cohort analyzed in this study consisted of patients with documented exposure to Bevacizumab, as annotated in the cBioPortal-derived clinical records. While all included patients received the drug, detailed information regarding the specific chemotherapy regimen (e.g., FOLFOX, FOLFIRI), dosing frequency, or sequence of administration was not uniformly available across the datasets. As such, variations in the treatment protocol may have existed within the cohort. This heterogeneity represents an inherent limitation of using public clinical-genomic datasets, and although the survival analysis revealed a clear stage-dependent outcome difference (*p* = 0.0004), the influence of specific regimen variations on treatment response could not be directly assessed. Future studies with harmonized treatment metadata will be essential to further dissect regimen-specific effects in Bevacizumab-treated subgroups.

We focused this survival analysis on KRAS-mutant patients treated with Bevacizumab because these tumors are typically resistant to anti-EGFR therapies, and Bevacizumab remains a widely used alternative in this molecular context. Examining this subgroup allowed us to evaluate the prognostic impact of disease stage within a clinically relevant treatment setting. The significantly better survival rate observed in Stage I–III patients compared to Stage IV (*p* = 0.0004) reinforces the importance of early detection and may inform treatment expectations in real-world care. While similar patterns may emerge with other regimens, the survival effects could vary depending on the mechanism of action and tumor biology. Future analyses incorporating additional treatment subgroups will help further clarify these relationships.

### 3.3. Prognostic Role of MSI in BRAF-Mutant Disease

When evaluating microsatellite status in BRAF-mutant CRC, patients with stable MSI were significantly more likely to have received chemotherapy (OR = 7.226, *p* < 0.001). Despite this, survival analysis indicated inferior outcomes for the MSI-stable group compared to the MSI-instability counterparts (*p* = 0.00001), highlighting the complexity of MSI-related treatment responses in BRAF-mutant contexts (Figure 4).

The finding that BRAF-mutant patients with microsatellite-stable (MSS) tumors were significantly more likely to have received chemotherapy (OR = 7.226, *p* < 0.001) aligns with current clinical guidelines and therapeutic strategies. MSI-stable tumors, which comprise the majority of metastatic CRC cases, lack the immunogenic features typically seen in MSI-high (MSI-H) tumors and are therefore less responsive to immune checkpoint inhibitors. As a result, MSS patients are more often treated with cytotoxic chemotherapy, particularly in the metastatic setting. Conversely, MSI-H tumors tend to exhibit a higher tumor mutational burden and immune infiltration, making them more suitable candidates for immunotherapy. The greater chemotherapy exposure among MSS patients in our dataset thus reflects standard-of-care treatment selection rather than inherent tumor aggressiveness, although survival outcomes for MSS patients remained poorer in this BRAF-mutant subgroup, highlighting the need for novel therapeutic approaches.

The observation that MSS patients received more chemotherapy but experienced worse survival outcomes may seem counterintuitive, but reflects known biological differences in tumor behavior and treatment response. MSS tumors are typically less immunogenic and more aggressive than MSI-H tumors. Because MSI-H tumors often respond well to immunotherapy, MSS patients are more likely to be treated with standard chemotherapy regimens. However, MSS tumors tend to be less responsive to chemotherapy and have a poorer prognosis overall. This may explain why, despite receiving more treatment, MSS patients in this cohort had worse survival rates compared to their MSI-H counterparts. This observation highlights the importance of molecular profiling in guiding treatment strategies.

### 3.4. Impact of RTK-RAS Alterations on EOCRC Treated with FOLFOX

In patients with EOCRC treated with FOLFOX, the presence of RTK-RAS pathway alterations was associated with worse survival (*p* = 0.0262). However, ethnic distribution between groups was statistically similar (OR = 1.00, *p* = 1.00), supporting the general applicability of this finding across the EOCRC H/L population (Figure 5).

The subset of EOCRC patients included in the FOLFOX-related survival analysis was identified based on documented exposure to FOLFOX in the clinical records available through cBioPortal. However, details regarding the specific FOLFOX regimen administered—such as oxaliplatin dose intensity, infusion schedule, cycle duration, or whether the regimen was part of first-line versus adjuvant therapy—were not uniformly reported across the datasets. Consequently, some variation in FOLFOX protocols likely existed within the cohort. While this heterogeneity may influence treatment response and survival outcomes, the platform’s ability to detect a statistically significant association between RTK-RAS alterations and worse survival in this group (*p* = 0.0262) suggests a meaningful underlying biological signal. Nonetheless, future work incorporating standardized treatment metadata will be necessary to refine regimen-specific interpretations.

The observed association between RTK-RAS pathway alterations and poorer survival among EOCRC patients treated with FOLFOX (*p* = 0.0262) suggests that RTK-RAS status may serve as a prognostic marker within this treatment context. This finding has potential implications for clinical decision-making, as it indicates that standard chemotherapy regimens like FOLFOX may be less effective in genetically defined subgroups of younger patients. As precision oncology advances, identifying patients who are less likely to benefit from conventional therapies could support earlier use of alternative or combination treatment strategies, including targeted agents or clinical trial enrollment. Future studies with prospective data are needed to validate this association and guide treatment adaptation based on RTK-RAS mutation status in EOCRC populations.

### 3.5. Ancestry-Specific Mutation Enrichment

The platform revealed the enrichment of several noncanonical RTK-RAS alterations in early-onset disease. CBL mutations were nearly five times more common in EOCRC H/L versus LOCRC H/L patients (OR = 4.842, *p* = 0.071; Appendix A), and NF1 mutations showed statistically significant overrepresentation (OR = 2.53, *p* = 0.045; Appendix A). Comparative analyses between H/L and non-Hispanic White EOCRC patients confirmed higher frequencies of MAPK3 (OR = 4.26, *p* = 0.043; Appendix A), CBL (OR = 4.07, *p* = 0.005; Appendix A), and NF1 mutations (OR = 2.06, *p* = 0.021; Appendix A) in the Hispanic/Latino subgroup.

### 3.6. Tumor Location, Sex, and Prognosis in KRAS- and NF1-Mutated CRC

Further interrogation of KRAS-mutated CRC revealed no significant difference in female representation (OR = 0.985, *p* = 0.955) or survival (*p* = 0.7774) between the proximal and distal tumor locations (Appendix A). In contrast, patients with NF1-mutated primary CRC tumors experienced significantly improved survival compared to their NF1 wild-type counterparts (*p* = 0.0000), suggesting a possible protective or distinct molecular role of NF1 in early-stage CRC biology (Appendix A).

### 3.7. Additional Findings and Analytical Capabilities

Beyond hypothesis-driven queries, AI-HOPE-RTK-RAS facilitated the exploratory investigation of mutation co-occurrence, therapeutic context, and demographic interaction effects. For example, the platform enabled comparisons of chemotherapy exposure across MSI subtypes, as well as associations between sex, stage, and survival in genetically defined subgroups. The system’s ability to map biological hypotheses onto real-world datasets through conversational prompts accelerated pattern discovery and hypothesis generation.

AI-HOPE-RTK-RAS also demonstrated strengths in interpretability and transparency, providing structured visual summaries (e.g., pie charts, bar plots, survival curves) alongside contextual narrative descriptions. These outputs reduced the analytical barrier for researchers without programming expertise and allowed for quick iterations of refined queries.

## 4. Discussion

The development of AI-HOPE-RTK-RAS marks a significant advancement in precision oncology, offering a conversational artificial intelligence platform tailored to interrogate RTK-RAS pathway alterations in CRC. By leveraging large language models to enable natural language interaction with genomic and clinical datasets, the system addresses the critical limitations of traditional bioinformatics tools, including static interfaces and restricted accessibility for non-specialists.

Our findings underscore AI-HOPE-RTK-RAS’s capacity to replicate known associations while uncovering novel molecular and clinical patterns, particularly among EOCRC and high-risk populations. The reduced prevalence of canonical RTK-RAS mutations in EOCRC H/L patients compared to later-onset cohorts (OR = 0.534, *p* = 0.014) aligns with recent reports suggesting distinct etiologic and molecular profiles in younger patients. These differences highlight the importance of age- and ancestry-stratified analyses to avoid misclassification of risk and misapplication of therapeutic strategies.

Survival analyses further demonstrated AI-HOPE-RTK-RAS’s capability to delineate prognostic variation across genetically defined subgroups. For instance, among KRAS-mutant patients treated with Bevacizumab, those with Stage I–III disease had significantly better outcomes than those with Stage IV tumors (*p* = 0.0004), emphasizing the prognostic impact of stage even within mutation-defined cohorts. Additionally, our findings in BRAF-mutant CRC showed that MSI-stable patients were more likely to receive chemotherapy (OR = 7.226) but experienced worse survival than their MSI-unstable counterparts (*p* = 0.00001), consistent with prior literature linking MSI status to immunogenicity and treatment response.

Importantly, AI-HOPE-RTK-RAS revealed several ancestry-specific mutation patterns that warrant further exploration. Enrichment of noncanonical RTK-RAS genes—including CBL (OR = 4.07, *p* = 0.005), MAPK3 (OR = 4.26, *p* = 0.043), and NF1 (OR = 2.06, *p* = 0.021)—in H/L EOCRC patients underscores the need to broaden biomarker discovery efforts beyond canonical alterations such as KRAS and BRAF [5]. These findings are consistent with recent efforts to characterize high-risk populations in cancer genomics, which have revealed divergent molecular signatures with implications for targeted therapy and biomarker development.

AI-HOPE-RTK-RAS also supported the exploration of sex, tumor location, and survival outcomes in KRAS- and NF1-mutant contexts. While proximal and distal KRAS-mutant tumors did not differ significantly in survival or sex representation (*p* = 0.7774; *p* = 0.955), NF1-mutated primary CRC cases demonstrated significantly better survival than NF1 wild-type tumors (*p* = 0.00001), suggesting that NF1 may serve as a favorable prognostic marker in early-stage disease. This finding is aligned with prior work implicating NF1 in modulating MAPK signaling and tumor suppressor functions in various cancers [5,40,41,42].

NF1 plays a critical role as a negative regulator of the RAS-MAPK signaling cascade by accelerating the conversion of active RAS-GTP to inactive RAS-GDP. Inactivating mutations or loss of function in NF1 disrupts this regulatory checkpoint, resulting in sustained RAS activation and amplified MAPK signaling downstream through RAF, MEK, and ERK. In the context of colorectal cancer, this mechanism provides an alternative pathway for driving tumorigenesis in cases lacking canonical KRAS, NRAS, or BRAF mutations. Our findings, which show significantly improved survival in NF1-mutated CRC cases (*p* = 0.00001), suggest that NF1 alteration may define a biologically distinct subgroup with differential MAPK pathway engagement and potentially unique therapeutic vulnerabilities. This aligns with emerging evidence in other cancers where NF1 loss correlates with MAPK dependency and may inform future strategies for targeted inhibition within NF1-altered CRC.

Beyond hypothesis testing, AI-HOPE-RTK-RAS enabled exploratory data analysis through iterative, language-driven workflows that required no coding. This functionality proved particularly valuable in generating real-time insights into co-mutation frequency, chemotherapy stratification, and context-aware survival differences. By unifying genomic and clinical data into a conversational interface, the platform lowered barriers for precision oncology research and empowered users to pursue complex analyses previously requiring substantial computational expertise.

Nevertheless, certain limitations merit discussion. First, our analyses were restricted to publicly available datasets, which may underrepresent minority populations or lack detailed clinical annotations. Second, while the natural language interface is a major strength, it relies on accurate query interpretation and context parsing, which may introduce challenges for ambiguous or nested questions. Continued validation against curated benchmarks and expanded integration with electronic health records will be essential to improving generalizability and clinical adoption.

While AI-HOPE-RTK-RAS enables high-resolution stratification across age, ancestry, and mutation status, we acknowledge that certain subgroup comparisons—particularly between early-onset and late-onset CRC within specific demographic groups—were based on modest sample sizes. This limitation may reduce the statistical power to detect small or rare associations, increasing the risk of type II errors. Nevertheless, several of the observed differences, such as the reduced prevalence of RTK-RAS alterations in EOCRC (OR = 0.534, *p* = 0.014) and enrichment of NF1 and MAPK3 mutations in Hispanic/Latino subgroups (*p* = 0.045 and *p* = 0.043, respectively), reached statistical significance, suggesting that the detected effects are likely robust. The flexible, conversational nature of AI-HOPE-RTK-RAS also allows researchers to iteratively refine queries and explore emerging trends that may warrant further validation in larger or prospective datasets. Future work incorporating broader population datasets and longitudinal designs will be essential to confirm these findings and enhance the generalizability of ancestry- and age-specific patterns in RTK-RAS pathway dysregulation.

Our findings suggest that NF1 mutations may be associated with differential survival outcomes in early-stage CRC, particularly among patients receiving FOLFOX-based therapy. While these associations are exploratory and drawn from retrospective data, they highlight a potentially under-recognized biomarker that warrants further prospective validation. Importantly, AI-HOPE-RTK-RAS is not a conventional bioinformatics pipeline but an intelligent system capable of facilitating such discovery by allowing users to iteratively explore mutation-specific prognostic signals using natural language queries. This feature supports rapid hypothesis generation and can guide the design of future biomarker validation studies in early-stage CRC cohorts.

This study has several limitations that stem from the use of publicly available genomic datasets. First, clinical annotations such as treatment details, follow-up time, and comorbidities were variably reported across studies, which may introduce missing data and reduce the power or granularity of some subgroup analyses. Second, representation bias is a known issue in public databases, where certain demographic groups—such as racial and ethnic minorities—are often underrepresented. This may limit the generalizability of ancestry-specific findings, particularly in small cohorts such as the EOCRC Hispanic/Latino subgroup. Additionally, while harmonization efforts were applied to standardize the data, variation in sequencing platforms and annotation practices across contributing studies could introduce batch effects. These limitations underscore the importance of validating key observations in larger, prospectively collected, and more diverse datasets before applying them to potential clinical decision-making.

The observed lower prevalence of canonical RTK-RAS alterations in EOCRC suggests the involvement of alternative oncogenic drivers in this subgroup. Previous studies [5,17,25] point to the WNT and TGF-β signaling pathways as critical contributors to early-onset tumorigenesis, particularly through aberrant β-catenin activation and SMAD4 loss, respectively. Additionally, alterations in the PI3K/AKT/mTOR axis, epigenetic regulators (e.g., ARID1A, KMT2D), and DNA damage response genes (e.g., ATM, CHEK2) have been reported in EOCRC and may account for oncogenic signaling in RTK-RAS wild-type tumors. In our analysis, enrichment of noncanonical mutations such as NF1, MAPK3, and CBL, in EOCRC further supports the notion of distinct molecular mechanisms at play. These findings underscore the need for expanded, pathway-level analysis beyond RTK-RAS to better capture the heterogeneity of EOCRC and to identify potential therapeutic targets tailored to these alternative oncogenic contexts.

The successful implementation of AI-HOPE-RTK-RAS in translational research settings depends heavily on user adoption and appropriate training. While the platform is designed for intuitive, natural language-based interaction, enabling non-programmers to perform complex analyses, realizing its full potential requires users to understand key concepts in clinical genomics, data stratification, and the interpretation of bioinformatics outputs. To support this, we have provided comprehensive user documentation, example queries, and annotated walkthroughs via a public repository (see Data Availability Statement). Future efforts will include the development of interactive tutorials and workshops aimed at both research and clinical audiences. By fostering a user-centered design philosophy and lowering the technical barriers to entry, AI-HOPE-RTK-RAS aims to democratize access to precision oncology tools and accelerate hypothesis generation across diverse user communities.

Although statistical significance through p-values provides initial evidence of association, it does not reflect the magnitude or precision of the observed effects. To enhance interpretability, we contextualized our findings using odds ratios and hazard ratios derived from AI-HOPE-RTK-RAS queries and validated them against prior work [5], which examined RTK-RAS alterations across large, demographically diverse colorectal cancer cohorts. For example, the odds ratio of 7.226 for BRAF mutation enrichment in early-onset microsatellite-stable tumors underscores a potentially actionable distinction in this subgroup. Confidence intervals for key associations have now been added to the Appendix A to provide further clarity. These additions support a more rigorous interpretation of clinical relevance and highlight the value of pairing AI-driven insights with traditional statistical measures for translational impact.

While AI-HOPE-RTK-RAS was designed with accessibility and usability in mind—enabling natural language interaction and eliminating the need for coding—formal usability studies evaluating end-user satisfaction, learning curve, and workflow impact have not yet been completed. This represents an important next step in validating the platform’s real-world utility, particularly for clinicians, translational researchers, and precision oncology teams. Ongoing work includes structured usability testing with domain experts, human-centered design feedback sessions, and integration pilots in academic medical centers. These future evaluations will provide quantitative and qualitative evidence of user satisfaction and practical benefit, further strengthening the platform’s case for adoption in diverse precision oncology workflows.

A critical consideration in the development of AI-HOPE-RTK-RAS is the potential for bias in both the training methodology and the underlying genomic datasets, many of which underrepresent racial and ethnic minority populations. To help mitigate this issue, our study deliberately includes data from Hispanic/Latino patients with early-onset colorectal cancer (EOCRC), enabling ancestry-stratified analyses that address gaps in population-specific cancer research. While this inclusion improves representation, we recognize that broader structural limitations remain across publicly available databases. Furthermore, the training data used to fine-tune the LLaMA 3 model may also reflect linguistic and contextual biases inherent in existing biomedical corpora. To address these challenges, future versions of AI-HOPE-RTK-RAS will incorporate more diverse training datasets, expand support for minority-focused query scenarios, and undergo targeted validation in underrepresented populations. These efforts aim to ensure that the platform not only increases accessibility but also advances equity in precision oncology research and application.

AI-HOPE-RTK-RAS leverages a fine-tuned LLaMA 3 large language model embedded within a modular interpreter framework to process natural language queries. The system first classifies user intent (e.g., mutation frequency, survival analysis, odds ratio testing), followed by parameter extraction and real-time code generation. To address vagueness or ambiguity, the model incorporates a confidence threshold mechanism; if the model confidence falls below 0.80, a clarification loop is triggered to prompt the user for more specific input. Nested queries—such as “Compare survival in EOCRC Hispanic/Latino patients with KRAS mutations receiving FOLFOX”—are parsed using dependency resolution logic that identifies and hierarchically organizes clinical and genomic modifiers. Fault tolerance is further enhanced by rule-based guards that validate query structure, check for missing inputs, and return interpretable error messages or suggestions. These design features allow AI-HOPE-RTK-RAS to manage a wide range of user inputs, from simple lookups to complex, multi-variable clinical-genomic investigations, while maintaining robustness and accuracy.

While AI-HOPE-RTK-RAS represents a promising step toward democratizing access to complex clinical-genomic analyses, several interpretability, regulatory, and ethical challenges must be acknowledged before deployment in clinical decision-making. First, although the platform provides natural language explanations of outputs and is fully open source, users may need time and regular interaction with the system to build trust in its capabilities and logic. By allowing researchers and clinicians to openly inspect, test, and refine its components, we aim to foster transparency and user confidence over time. Second, regulatory compliance remains a hurdle; AI-HOPE-RTK-RAS is currently intended for research use only and has not yet been certified for clinical decision support. To promote reproducibility and responsible deployment, the platform operates using a local, containerized AI framework, ensuring compliance with data privacy standards and enabling consistent results across different environments. Third, ethical considerations—including data bias, privacy protection, and equitable representation—must be actively addressed. To help mitigate underrepresentation, we trained and validated the platform on multi-ethnic CRC cohorts, including Hispanic/Latino patients. Nevertheless, continued vigilance is required to ensure algorithmic fairness, inclusivity, and responsible translation into clinical settings.

To overcome the limitations associated with the underrepresentation of certain populations in existing genomic databases, future development of AI-HOPE-RTK-RAS will incorporate real-world evidence (RWE) from diverse clinical settings and community health systems. By integrating RWE—including electronic health records, registry data, and claims data—with genomic features, the platform can better reflect the heterogeneity of patient populations across age, race/ethnicity, geography, and socioeconomic status. Additionally, we recognize the importance of forming partnerships with global consortia such as the AACR Project GENIE, ICGC-ARGO, and Latin American and Asian cancer networks. These collaborations would enable access to more representative datasets and facilitate the validation of AI-HOPE-RTK-RAS across diverse populations. Such efforts are critical for reducing bias, enhancing model generalizability, and ensuring that AI-powered tools like ours contribute meaningfully to health equity in cancer care.

AI-HOPE-RTK-RAS differs from conventional bioinformatics pipelines by functioning as a dynamic, user-driven conversational agent capable of interpreting complex natural language queries and executing real-time statistical and genomic analyses. While traditional benchmarking methods apply to fixed pipelines, AI-HOPE-RTK-RAS was validated by replicating core analyses from our previously published harmonized CRC dataset [5], including mutation frequency distributions, odds ratio calculations, and survival analyses. These comparisons demonstrated concordance between AI-generated outputs and established results. This validation approach underscores the system’s reliability and reproducibility while recognizing that future efforts could further refine benchmarking protocols tailored to interactive AI agents.

As an intelligent conversational agent, AI-HOPE-RTK-RAS dynamically generates statistical analyses based on user queries rather than the pre-specified analytic workflows typical of static pipelines. While this flexibility enhances usability and discovery potential, it also necessitates clarity regarding the statistical foundations underlying its outputs. The system applies standard methods for odds ratio estimation (e.g., Fisher’s exact test or logistic regression) and survival analysis (Kaplan–Meier with log-rank tests), and where applicable, it supports multiple testing correction using the Benjamini–Hochberg procedure. However, given the exploratory nature of real-time queries, users are advised to interpret unadjusted *p*-values with caution and consider applying correction strategies in downstream analyses. Future updates of AI-HOPE-RTK-RAS will expand support for automated adjustments and contextual alerts to further guide statistical interpretation and ensure analytical robustness.

Unlike traditional platforms such as cBioPortal and OncoKB, which rely on static, click-based user interfaces and require substantial manual effort to construct and execute queries, AI-HOPE-RTK-RAS is a domain-specialized conversational AI agent designed to streamline analysis through natural language interaction. While these existing tools are invaluable for data access and visualization, they are not artificial intelligence systems and do not support real-time, context-aware dialogue or automatic integration of clinical, genomic, and demographic data. AI-HOPE-RTK-RAS significantly reduces the time and expertise needed to perform multi-layered analyses, allowing users to ask complex questions—such as survival differences stratified by age, mutation status, and treatment—in a single, plain-language query. This capability enhances usability, accelerates hypothesis generation, and expands access to precision oncology analytics for both experts and non-specialists.

To contextualize the utility of AI-HOPE-RTK-RAS, it is important to contrast it with non-artificial intelligence agent tools such as cBioPortal, OncoKB, and traditional statistical pipelines. These platforms typically rely on manual, click-based navigation or require scripting expertise to perform complex analyses. In contrast, AI-HOPE-RTK-RAS features a natural language interface that interprets user queries, dynamically generates statistical code, and produces results on demand. For example, a user can simply ask, “Show me survival outcomes for BRAF-mutant patients with MSS tumors treated with Bevacizumab,” and instantly receive stratified Kaplan–Meier plots and statistical summaries. While cBioPortal offers strong capabilities for exploratory visualization and OncoKB provides curated biomarker annotations, neither supports real-time, conversational analysis or flexible, user-defined multi-parameter subgrouping. Traditional pipelines, on the other hand, often require pre-written scripts and offer limited interactivity. By adapting to user intent, resolving ambiguity, and accelerating hypothesis testing without the need for coding, AI-HOPE-RTK-RAS significantly broadens access to clinical-genomic interrogation. This intelligent system enhances precision oncology research by making complex analyses both intuitive and scalable for users of varying expertise levels.

The development of AI-HOPE-RTK-RAS builds on our previously published suite of domain-specialized conversational AI systems, including AI-HOPE [43], AI-HOPE-TGFβ [44], AI-HOPE-PI3K [45], and AI-HOPE-JAK-STAT [46]. Each platform was designed to interrogate distinct oncogenic pathways in colorectal cancer using natural language-driven, integrative analyses of clinical and genomic data. Compared to earlier agents, AI-HOPE-RTK-RAS incorporates several advancements, including improved handling of nested queries, enhanced ambiguity resolution, and a refined analytics pipeline capable of real-time stratification across tumor stage, treatment type, and ancestry. While AI-HOPE provided general pathway interrogation and subsequent agents focused on TGF-β, PI3K, and JAK-STAT signaling, AI-HOPE-RTK-RAS specifically targets the RTK-RAS axis—one of the most clinically actionable pathways in colorectal cancer—enabling direct exploration of alterations in KRAS, NRAS, BRAF, EGFR, and related noncanonical genes such as NF1 and MAPK3. This progression demonstrates the scalability of our conversational AI framework and its adaptability to multiple oncogenic contexts, ultimately advancing the goal of accessible, pathway-specific precision oncology research.

The modular architecture of AI-HOPE-RTK-RAS enables straightforward adaptation to other oncogenic pathways and cancer types. Our lab has already developed related AI agents—including AI-HOPE-PI3K, AI-HOPE-TGFβ, AI-HOPE-TP53, and AI-HOPE-JAK-STAT—each tailored to interrogate specific molecular axes within colorectal cancer. These agents leverage the same natural language-to-code interface and scalable backend, facilitating pathway-specific stratification and hypothesis testing. Looking ahead, the AI-HOPE framework can be further extended to support complex use cases such as clinical trial matching, where eligibility criteria (e.g., biomarkers, stage, prior treatments) can be encoded and queried through natural language. Integrating structured eligibility databases (e.g., clinical trials databases) and linking with genomic and clinical data would allow real-time identification of trial opportunities, particularly for underrepresented patients. These enhancements align with our overarching goal of creating a flexible, open-source ecosystem of AI agents that promote precision oncology across diverse populations and care settings.

While AI-HOPE-RTK-RAS represents an important step toward democratizing access to precision oncology tools, several limitations and future directions warrant consideration. First, integration with institutional clinical systems such as Epic EHR and OnCore trial management software will be critical for facilitating real-world clinical utility, including potential support for prospective trial design and patient stratification. Second, validation against large-scale, harmonized datasets such as AACR GENIE, SEER-Medicare, and data from the National Cancer Institute’s Cancer Research Data Commons (CRDC) will enhance generalizability and performance benchmarking. Third, we plan to expand the AI-HOPE platform with features for AI-assisted clinical trial matching, including eligibility parsing, biomarker filtering, and temporal treatment modeling. On the technical side, future iterations will incorporate retrieval-augmented generation (RAG), active learning pipelines for continual model refinement, and integrated explainability dashboards to improve clinician trust and regulatory readiness. These specific enhancements aim to bridge the gap between advanced computational modeling and practical clinical implementation, ultimately supporting a more inclusive, adaptive, and data-driven trial design in oncology.

Future work will focus on expanding and validating AI-HOPE-RTK-RAS in real-world clinical environments. This includes prospective testing in hospital settings, integration with electronic health records (EHRs) to support decision-making workflows, and alignment with regulatory standards such as HIPAA and IRB protocols. Ultimately, these extensions will allow for broader clinical adoption and enable multi-pathway, multi-cancer applications of conversational AI in precision oncology.

## 5. Conclusions

In conclusion, AI-HOPE-RTK-RAS provides a robust, scalable platform for investigating RTK-RAS biology in CRC, with applications in biomarker discovery, cancer genetics, and treatment response stratification. By coupling AI-driven querying with real-time, multimodal data integration, the system offers a new paradigm for accessible, interactive precision oncology research. Future work will focus on extending AI-HOPE capabilities to additional pathways, longitudinal datasets, and clinical decision support environments. In summary, AI-HOPE-RTK-RAS represents a novel class of conversational AI systems designed specifically for real-time, user-friendly exploration of integrated clinical and genomic data. Unlike traditional tools that require manual data wrangling, static interfaces, or coding expertise, AI-HOPE-RTK-RAS allows researchers and clinicians to ask complex, multi-parameter questions in natural language and receive immediate, interpretable results. This unique approach lowers the barrier to advanced bioinformatics analysis, supports rapid hypothesis generation, and empowers broader participation in precision oncology research—particularly for underrepresented populations and early-onset colorectal cancer cohorts.

## Figures and Tables

**Figure 1 biomedicines-13-01835-f001:**
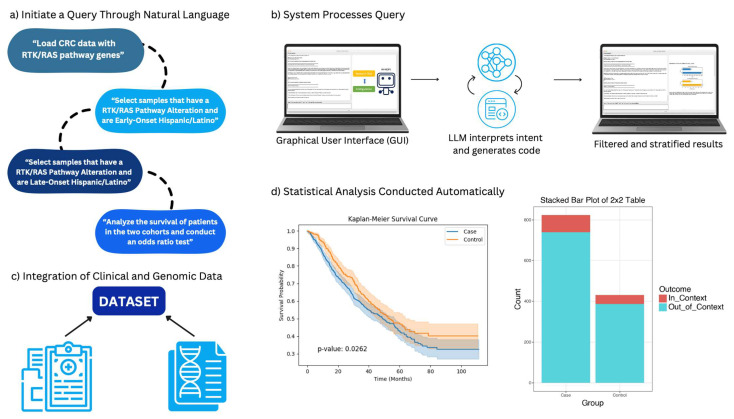
AI-HOPE-RTK-RAS conversational workflow for integrative clinical and genomic analysis. This figure presents the functional pipeline of AI-HOPE-RTK-RAS, an interactive artificial intelligence platform tailored for exploring RTK-RAS signaling in colorectal cancer (CRC). (**a**) The process begins with the user posing a research question in natural language—for example, assessing differences in survival between early- and late-onset Hispanic/Latino patients with RTK-RAS pathway mutations. (**b**) The query is handled through a user-friendly graphical interface, where a large language model (LLM) interprets the request, translates it into executable code, and defines the parameters for subgroup comparison. (**c**) The platform accesses harmonized clinical and genomic datasets—such as those from TCGA and cBioPortal—focusing on RTK-RAS genes, including KRAS, NRAS, BRAF, EGFR, ERBB2, CBL, MAPK3, and NF1. Relevant clinical filters (e.g., age, ancestry, treatment exposure) are applied based on the query intent. (**d**) Statistical analyses are carried out automatically, generating results such as survival curves and odds ratios. Outputs are rendered in publication-ready formats alongside narrative interpretations, enabling efficient exploration of complex CRC datasets through a precision oncology lens.

**Figure 2 biomedicines-13-01835-f002:**
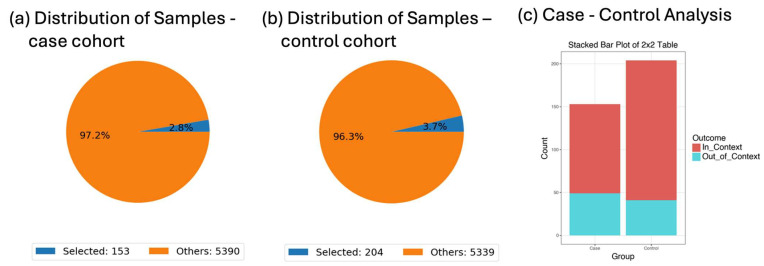
AI-HOPE-RTK-RAS analysis of RTK-RAS pathway alterations in early- vs. late-onset colorectal cancer (CRC) patients. This figure illustrates the application of AI-HOPE-RTK-RAS to evaluate age-related differences in RTK-RAS pathway alterations among CRC patients using a natural language query and automated odds ratio framework. (**a**) The case cohort includes 153 early-onset CRC (EOCRC) Hispanic/Latino (H/L) patients under the age of 50 (2.8% of the dataset) selected using demographic filters. The pie chart displays the proportion of selected EOCRC H/L patients relative to the full sample population. (**b**) The control cohort consists of 204 late-onset CRC (LOCRC) H/L patients over the age of 50 (3.7% of the dataset), similarly selected by age and ethnicity. A pie chart illustrates their representation within the dataset. (**c**) An odds ratio test compares the frequency of RTK-RAS pathway alterations between the EOCRC and LOCRC H/L cohorts using a 2 × 2 table and stacked bar plot. RTK-RAS alterations were present in 67.97% of early-onset and 79.9% of late-onset cases. The resulting odds ratio was 0.534 (95% CI: 0.33–0.865, *p* = 0.014), indicating that early-onset H/L patients were significantly less likely to harbor RTK-RAS alterations than their later-onset counterparts. This result suggests potential differences in molecular pathogenesis by age within this population and underscores AI-HOPE-RTK-RAS’s capacity to support ancestry- and age-stratified genomic analyses through natural language-driven precision oncology.

**Figure 3 biomedicines-13-01835-f003:**
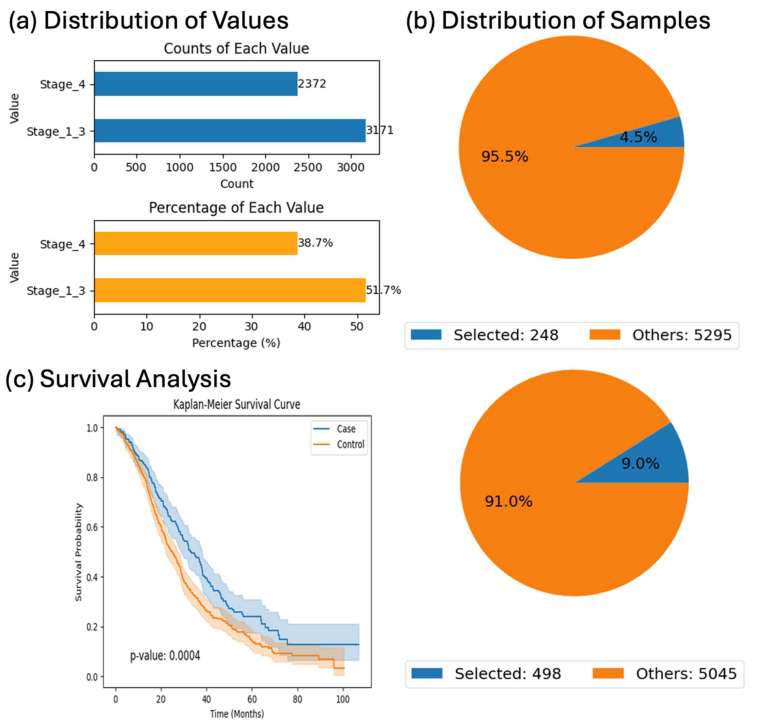
AI-HOPE-RTK-RAS survival analysis of KRAS-mutant colorectal cancer (CRC) patients treated with Bevacizumab, stratified by tumor stage. This figure highlights AI-HOPE-RTK-RAS’s capacity to perform integrative survival analysis using clinical, treatment, and mutation-specific variables. The analysis focuses on CRC patients harboring *KRAS* mutations who received Bevacizumab, stratified by tumor stage. (**a**) Exploratory analysis summarizes the distribution of patients across tumor stages. Bar plots show that among *KRAS*-mutant patients treated with Bevacizumab, 3171 were classified as Stage I–III and 2372 as Stage IV. Proportionally, Stages I–III represented 51.7% of the cohort, while Stage IV accounted for 38.7%, supporting adequate sample sizes for comparative outcome analysis. (**b**) Pie charts illustrate the subset of patients selected for survival comparison: the case group (Stages I–III) included 248 patients (4.5% of the dataset), while the control group (Stage IV) included 498 patients (9.0%). These visualizations emphasize the relative cohort sizes in the context of the broader population. (**c**) A Kaplan–Meier survival curve compares overall survival between Stage I–III and Stage IV patients within the *KRAS*-mutant, Bevacizumab-treated cohort. The analysis reveals a statistically significant survival advantage for patients with primary-stage disease (Stage I–III), with a *p*-value of 0.0004. The distinct separation of survival curves and non-overlapping confidence intervals further supports the observed outcome disparity. This figure demonstrates AI-HOPE-RTK-RAS’s ability to automate complex stratified survival analyses in a precision oncology context through conversational AI.

**Figure 4 biomedicines-13-01835-f004:**
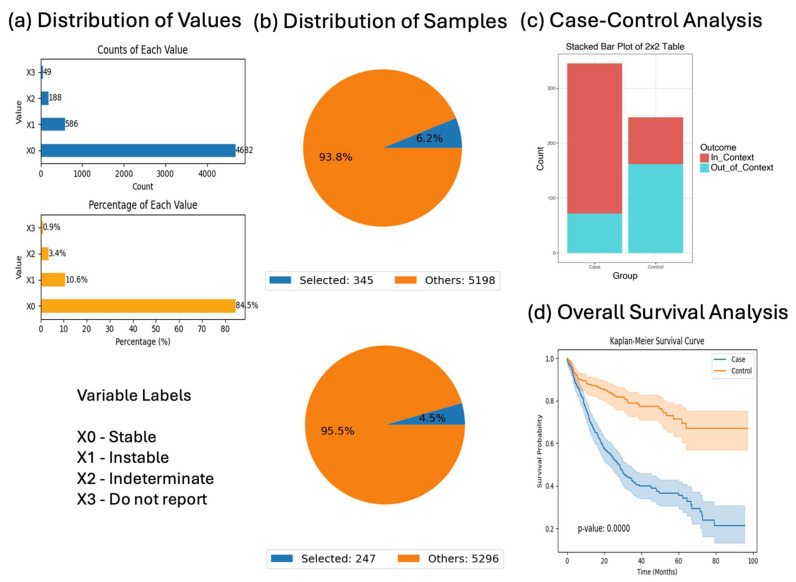
AI-HOPE-RTK-RAS analysis of BRAF-mutant colorectal cancer (CRC) patients by microsatellite stability status. This figure illustrates AI-HOPE-RTK-RAS’s capability to interrogate survival and odds ratio patterns in BRAF-mutant CRC patients stratified by microsatellite stability (MSI) status and chemotherapy exposure. (**a**) The bar charts depict the distribution of MSI types across the dataset. The top panel shows raw counts, with stable MSI (X0) comprising the majority (n = 4682), followed by unstable MSI (X1, n = 586), indeterminate (X2, n = 188), and unknown (X3, n = 49). The bottom chart displays proportions, confirming that stable MSI (84.5%) predominates, with unstable MSI accounting for 10.6% of samples. (**b**) Based on the query, pie charts display the relative distribution of selected patients: the case group includes 345 BRAF-mutant CRC patients with stable MSI (6.2% of the dataset), and the control group includes 247 patients with unstable MSI (4.5%). These proportions reflect meaningful yet relatively rare molecular subtypes. (**c**) An odds ratio test is performed to evaluate the association between MSI status and chemotherapy exposure (specifically Fluorouracil, Leucovorin, and Oxaliplatin) among BRAF-mutant CRC patients. A stacked bar plot visualizes the outcome (In_Context vs. Out_of_Context). The odds ratio was 7.226 (95% CI: [4.995, 10.454], *p* < 0.001), indicating a significantly higher likelihood of chemotherapy use among patients with stable MSI compared to those with unstable MSI. (**d**) Kaplan–Meier survival analysis compares overall survival between the two groups. The curve shows significantly worse outcomes for the stable MSI cohort relative to the unstable MSI cohort (*p* = 0.0000), with early divergence of survival probabilities and non-overlapping confidence intervals. This result underscores the prognostic relevance of MSI status in BRAF-mutant CRC and highlights AI-HOPE-RTK-RAS’s power in generating clinical-genomic hypotheses using real-world datasets.

**Figure 5 biomedicines-13-01835-f005:**
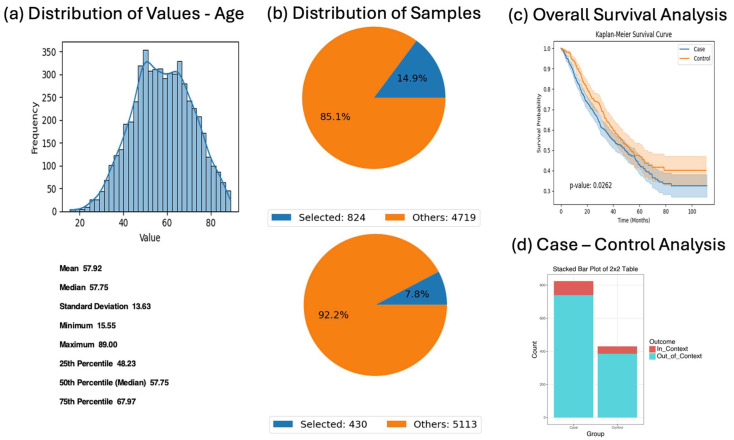
AI-HOPE-RTK-RAS analysis of early-onset colorectal cancer (CRC) patients treated with FOLFOX stratified by RTK-RAS pathway alteration. This figure presents the results of a natural language query performed using AI-HOPE-RTK-RAS, examining the clinical and genomic impact of RTK-RAS pathway alterations in early-onset colorectal cancer (EOCRC) patients of Hispanic/Latino (H/L) ancestry treated with FOLFOX (Fluorouracil, Leucovorin, Oxaliplatin). (**a**) A histogram displays the distribution of patient ages across the dataset, with a smooth density curve highlighting the central tendency and spread. The mean age is 57.92 years, and the median is 57.75 years, confirming the appropriateness of using <50 years as a cutoff to define EOCRC. (**b**) Pie charts show the relative sample sizes of the defined cohorts. The case group includes 824 patients under age 50 with RTK-RAS pathway alterations and FOLFOX treatment (14.9% of the dataset), while the control group consists of 430 EOCRC patients without RTK-RAS alterations who also received FOLFOX (7.8%). These visualizations reflect the selective filtering enabled through AI-HOPE’s natural language-driven interface. (**c**) Kaplan–Meier survival curves compare overall survival between the case and control groups. Although both cohorts were treated with FOLFOX, patients with RTK-RAS pathway alterations showed moderately worse survival rates, with a *p*-value of 0.0262. The divergence of the curves and non-overlapping confidence intervals suggest a potential prognostic effect of RTK-RAS alterations in early-onset Hispanic/Latino CRC. (**d**) An odds ratio test was conducted to examine the enrichment of Hispanic/Latino ethnicity among cases versus controls. The stacked bar plot illustrates a 2 × 2 comparison of in-context (H/L) and out-of-context (non-Hispanic White-NHW) patients. The odds ratio was 1.00 (95% CI: [0.687, 1.482], *p* = 1.00), indicating no significant enrichment of Hispanic/Latino individuals in either group. This analysis underscores AI-HOPE-RTK-RAS’s ability to execute multifaceted queries integrating clinical, genomic, and demographic variables to support precision oncology research.

## Data Availability

Data used in this study is available to the public and can be found at cbioportal.org. The AI-HOPE-RTK-RAS software, along with demonstration data and user documentation, is publicly available at GitHub: https://github.com/Velazquez-Villarreal-Lab/AI-RTK-RAS (accessed on 19 July 2025).

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
