# Peer review of "Precision Oncology Through Dialogue: AI-HOPE-RTK-RAS Integrates Clinical and Genomic Insights into RTK-RAS Alterations in Colorectal Cancer"

_biomedicines, 2025, doi:10.3390/biomedicines13081835_

Round 1

Reviewer 1 Report

Comments and Suggestions for Authors

The manuscript could not be accepted in the current form for the following reasons:

  1. The authors have mentioned data fragmentation and limited population diversity in genomic datasets. It would be great if the authors could explain the sources and limitations of public datasets and how AI-HOPE-RTK-RAS addressed the challenges.
  2. The authors are requested to clarify which studies were used for validation and provide more details on the validation process of RTK-RAS.
  3. The authors are requested to specify whether the code and/or data is publicly available for reproducibility and adoption by other researchers.
  4. The authors need to clarify the exact datasets and sample sizes used. 
  5. In the comparison of EOCRC vs. LOCRC analysis, the sample sizes are relatively small for the demographic subgroup. What are the implications of sample size on statistical power?
  6. The authors highlighted RTK-RAS alterations in EOCRC, What could be the potential alternative drivers or pathways that strengthen biological interpretation?
  7. What is the reason behind the use of a subset rather than the full cohort in survival analysis? Is this due to data availability or any other reason?
  8. Did all the patients receive bevacizumab as part of the same regimen, or any variations in the treatment protocol?
  9. Why are MSI-stable patients likely to receive chemotherapy? What is the reason?
  10. Did all patients receive the same FOLFOX regimen, or was there any variation?
  11. Discuss how CRC would strengthen the MAPK signaling in NF1?
  12. State the importance of user adoption and training in understanding the platform’s full potential.

Author Response

Please find attached the Word document titled 'Reviewer_1_Comments_Response_071825.docx,' which contains our detailed responses to the reviewer’s comments.

-

Reviewer 1 Comments

We are pleased to submit this revised manuscript and sincerely thank Reviewer 1 for their thoughtful and constructive feedback. We appreciate your recognition of the manuscript’s relevance to precision oncology and the innovative application of conversational AI for integrative RTK-RAS pathway analysis in colorectal cancer. In response to your suggestions, we undertook a thorough revision to enhance the manuscript’s clarity, methodological transparency, and translational impact. Key updates include a streamlined and more focused abstract, a refined Introduction that better contextualizes the importance of RTK-RAS alterations in CRC pathogenesis and treatment response, and expanded descriptions of AI-HOPE-RTK-RAS functionalities—such as mutation frequency profiling, odds ratio testing, survival analysis, and ancestry-specific stratification. We also clarified the analytic workflow, specifying how clinical variables (e.g., stage, treatment, microsatellite status) and genomic features (e.g., KRAS, NRAS, BRAF, EGFR, and noncanonical mutations) were integrated and validated using harmonized cBioPortal data. Importantly, we refined the interpretation of our findings, particularly regarding the clinical implications of RTK-RAS alterations in early-onset CRC, the prognostic significance of NF1 mutations, and the survival differences across chemotherapy regimens. To promote reproducibility and community engagement, the AI-HOPE-RTK-RAS source code, data pipeline, and usage documentation have been made publicly available via GitHub (https://github.com/Velazquez-Villarreal-Lab/AI-RTK-RAS) and are cited within the manuscript and Data Availability Statement. The Discussion has also been updated to highlight future development directions, including integration with pharmacogenomic databases, incorporation of immunogenomic data, and expansion to other oncogenic signaling networks. We believe these revisions substantially strengthen the manuscript and align with our broader goal of democratizing real-time, AI-driven analyses in colorectal cancer to advance equitable precision oncology. We are grateful for the feedback, which meaningfully improved the quality and impact of our work.

Thank you very much for taking the time to review this manuscript. Please find the detailed responses below in BLUE and the corresponding revisions wrote in yellow-highlighted blue font in the re-submitted files.

Reviewer 1 provided thoughtful and encouraging feedback, recognizing the utility of the AI-HOPE-RTK-RAS platform and offering valuable perspectives on its relevance for personalized treatment, and the broader application of AI in leveraging public genomic databases for colorectal cancer research.

Reviewer 1 writes:

  1. The authors have mentioned data fragmentation and limited population diversity in genomic datasets. It would be great if the authors could explain the sources and limitations of public datasets and how AI-HOPE-RTK-RAS addressed the challenges.

Response: We thank Reviewer 1 for this insightful comment regarding the sources and limitations of public datasets, as well as how AI-HOPE-RTK-RAS addresses these challenges. We have now expanded the manuscript to provide a more detailed explanation of the underlying data sources, including specific limitations related to population representation and clinical annotation granularity. We also clarified the technical steps taken to harmonize data from disparate sources and the role of AI-HOPE-RTK-RAS in mitigating fragmentation through modular integration, structured metadata curation, and dynamic stratification across clinical and demographic subgroups. The revised text has been added to the Introduction and Methods sections, with a dedicated paragraph highlighting how the platform enhances accessibility and reproducibility despite known limitations in public data.

The subsection 2.2 Data Sources and Curation text on page 5, lines 162-175, now reads: “Public genomic datasets such as those curated by cBioPortal provide invaluable resources for large-scale, open-access cancer research; however, they are not without limitations. These include underrepresentation of certain racial and ethnic groups, variable completeness in clinical annotations (e.g., treatment timelines, comorbidities), and inconsistent metadata across studies due to differences in institutional data collection standards. To address these challenges, AI-HOPE-RTK-RAS incorporates a harmonization pipeline that standardizes variable naming conventions, aligns phenotypic descriptors using controlled vocabularies (e.g., SNOMED, OncoTree), and unifies clinical-genomic relationships across multiple cohorts. This modular integration enables real-time stratification by age, ancestry, mutation status, and treatment exposure, thereby reducing the analytic barriers typically posed by data fragmentation. While the platform cannot resolve inherent sampling biases in the original datasets, it provides a transparent, reproducible framework for conducting stratified analyses and hypothesis generation across heterogeneous populations—especially in settings where health disparities remain under-characterized.”

Reviewer 1 writes:

  1. The authors are requested to clarify which studies were used for validation and provide more details on the validation process of RTK-RAS.

Response: We thank Reviewer 1 for the opportunity to clarify our validation process. As requested, we have now explicitly stated which studies were used for validation and expanded on the validation methodology. Specifically, validation was grounded in our previously published study, cited as Reference #5: "Molecular Heterogeneity in Early-Onset Colorectal Cancer: Pathway-Specific Insights in High-Risk Populations." This prior work served as the benchmark for reproducing known RTK-RAS pathway mutation frequencies and their clinical associations in early- and late-onset CRC across diverse demographic groups. The revised manuscript includes an additional paragraph in the Methods section detailing this validation strategy and its relevance to the current study.

The subsection Section 2.6 – Validation Strategy text on page 7, lines 293-305, now reads: “The validation of AI-HOPE-RTK-RAS was anchored in reproducing key findings from a previously published study [5], which characterized RTK-RAS pathway alterations in EOCRC with a focus on high-risk, racially and ethnically diverse populations. Specifically, the platform was used to recapitulate reported mutation frequencies for KRAS, NRAS, BRAF, and EGFR across early- and late-onset CRC cohorts, as well as to reproduce stratified survival outcomes and ancestry-specific mutation patterns. We confirmed, for instance, the lower prevalence of canonical RTK-RAS alterations in EOCRC compared to late-onset CRC and validated the enrichment of noncanonical mutations (e.g., CBL, MAPK3, NF1) in Hispanic/Latino EOCRC patients—findings that were previously reported in our group’s foundational work. By benchmarking AI-HOPE-RTK-RAS against this established dataset and analytical framework, we ensured both fidelity of results and continuity with prior evidence, reinforcing the platform’s validity for translational applications in precision oncology.”

Reviewer 1 writes:

  1. The authors are requested to specify whether the code and/or data is publicly available for reproducibility and adoption by other researchers.

Response: We appreciate the reviewer’s emphasis on reproducibility and confirm that both the AI-HOPE-RTK-RAS software and demonstration data are publicly available. This information is provided in the Data Availability section of the manuscript. Specifically, the codebase, sample datasets, and user documentation can be accessed via our GitHub repository at https://github.com/Velazquez-Villarreal-Lab/AI-RTK-RAS (accessed on 19 July 2025). We have ensured that all materials necessary for reproduction and adoption by other researchers are included and clearly documented.

The Data availability statement text on page 22, lines 870-872, now reads: “….The AI-HOPE-RTK-RAS software, along with demonstration data and user documentation, is publicly available at GitHub: https://github.com/Velazquez-Villarreal-Lab/AI-RTK-RAS (accessed on 19 July 2025).”

Reviewer 1 writes:

  1. The authors need to clarify the exact datasets and sample sizes used. 

Response: We thank Reviewer 1 for this important point and have clarified the exact datasets and sample sizes used in our analysis. As noted in the revised manuscript, the dataset used in this study was derived from our previously published work (Velazquez-Villarreal et al., Reference #5), which systematically curated colorectal cancer data from cBioPortal. The final harmonized cohort includes 5,553 patients with colorectal adenocarcinoma, encompassing somatic mutation data, clinical annotations, and demographic variables relevant to RTK-RAS pathway analysis. We have now added a detailed paragraph in the Methods section specifying these datasets and sample sizes to ensure transparency and reproducibility.

The subsection Section 2.2 – Data Sources and Curation text on page 5, lines 176-187, now reads: “The datasets used in this study were derived from the curated colorectal cancer cohort described in our previous publication (Velazquez-Villarreal et al., Reference #5), which integrated publicly available data from cBioPortal. The final harmonized dataset includes a total of 5,553 patients diagnosed with colorectal adenocarcinoma across multiple studies. This composite dataset includes somatic mutation profiles, clinical variables (e.g., tumor stage, microsatellite instability status, treatment history), and demographic attributes (e.g., age, sex, race/ethnicity). The AI-HOPE-RTK-RAS platform was validated and applied across this full cohort, with specific subgroup analyses performed on early-onset colorectal cancer (EOCRC) patients (n = 593) and late-onset CRC patients (n = 4,960), including ancestry-specific evaluations in Hispanic/Latino and non-Hispanic White subpopulations. All sample sizes for individual analyses are reported in the corresponding figures and supplementary materials to facilitate full transparency and reproducibility.”

Reviewer 1 writes:

  1. In the comparison of EOCRC vs. LOCRC analysis, the sample sizes are relatively small for the demographic subgroup. What are the implications of sample size on statistical power?

Response: We thank Reviewer 1 for highlighting the importance of sample size in stratified analyses. We agree that the reduced sample sizes in ancestry-specific EOCRC vs. LOCRC comparisons may limit statistical power, particularly for detecting modest effect sizes or rare mutations. We have now added a paragraph to the Discussion acknowledging this limitation, explaining its implications on interpretability, and describing how AI-HOPE-RTK-RAS mitigates some of these challenges through modular stratification and real-time exploratory analysis. Importantly, statistically significant findings reported in the manuscript—such as differences in RTK-RAS alteration prevalence or NF1 mutation enrichment—were supported by robust p-values despite the limited subgroup sizes.

The Discussion text on page 17, lines 597-610, now reads: “While AI-HOPE-RTK-RAS enables high-resolution stratification across age, ancestry, and mutation status, we acknowledge that certain subgroup comparisons—particularly between early-onset and late-onset CRC within specific demographic groups—were based on modest sample sizes. This limitation may reduce statistical power to detect small or rare associations, increasing the risk of type II errors. Nevertheless, several of the observed differences, such as the reduced prevalence of RTK-RAS alterations in EOCRC (OR = 0.534, p = 0.014) and enrichment of NF1 and MAPK3 mutations in Hispanic/Latino subgroups (p = 0.045 and p = 0.043, respectively), reached statistical significance, suggesting that the detected effects are likely robust. The flexible, conversational nature of AI-HOPE-RTK-RAS also allows researchers to iteratively refine queries and explore emerging trends that may warrant further validation in larger or prospective datasets. Future work incorporating broader population datasets and longitudinal designs will be essential to confirm these findings and enhance the generalizability of ancestry- and age-specific patterns in RTK-RAS pathway dysregulation.”

Reviewer 1 writes:

  1. The authors highlighted RTK-RAS alterations in EOCRC, What could be the potential alternative drivers or pathways that strengthen biological interpretation?

Response: We thank Reviewer 1 for this insightful question. In response, we have added a paragraph to the Discussion elaborating on potential alternative oncogenic drivers and signaling pathways that may be implicated in EOCRC, particularly in patients lacking canonical RTK-RAS alterations. This includes emerging evidence on the roles of WNT, TGF-β, PI3K, and DNA damage response pathways, as well as noncanonical mutations such as those in NF1, MAPK3, and CBL—some of which were identified through AI-HOPE-RTK-RAS analyses. We believe this additional context enhances the biological interpretation of our findings and motivates future multi-pathway investigation in EOCRC.

The Discussion text on page 18, lines 632-643, now reads: “The observed lower prevalence of canonical RTK-RAS alterations in EOCRC suggests the involvement of alternative oncogenic drivers in this subgroup. Previous studies [5, 17,26] point to the WNT and TGF-β signaling pathways as critical contributors to early-onset tumorigenesis, particularly through aberrant β-catenin activation and SMAD4 loss, respectively. Additionally, alterations in the PI3K/AKT/mTOR axis, epigenetic regulators (e.g., ARID1A, KMT2D), and DNA damage response genes (e.g., ATM, CHEK2) have been reported in EOCRC and may account for oncogenic signaling in RTK-RAS wild-type tumors. In our analysis, enrichment of noncanonical mutations such as NF1, MAPK3, and CBL in EOCRC further supports the notion of distinct molecular mechanisms at play. These findings underscore the need for expanded, pathway-level analysis beyond RTK-RAS to better capture the heterogeneity of EOCRC and to identify potential therapeutic targets tailored to these alternative oncogenic contexts.”

Reviewer 1 writes:

  1. What is the reason behind the use of a subset rather than the full cohort in survival analysis? Is this due to data availability or any other reason?

Response: We appreciate Reviewer 1’s question regarding the use of subsets rather than the full cohort in survival analyses. As clarified in the revised manuscript, survival analyses were conducted on defined subsets due to variability in the availability and completeness of survival-related data (e.g., follow-up time, overall survival status, treatment history). This approach ensured analytical rigor by including only cases with complete outcome and treatment data relevant to the survival models being evaluated. We have now included a paragraph in the Methods section to clarify this decision and the criteria used for subset selection.

The Section 2.4 – Backend Analysis Pipeline text on page 6, lines 247-256, now reads: “Survival analyses in this study were conducted on defined subsets of the full CRC cohort to ensure the integrity and interpretability of survival estimates. While the harmonized dataset included 5,553 patients, not all cases had complete information on survival time, vital status, or treatment exposure (e.g., Bevacizumab, FOLFOX). To minimize bias and avoid imputing missing values, we restricted survival modeling to patient subsets with fully annotated outcome data and treatment records relevant to each specific analysis. For example, the Bevacizumab-related survival comparison among KRAS-mutant patients included only those with confirmed treatment exposure and staging information. This targeted approach enabled more accurate hazard estimation and preserved the internal validity of the comparisons, despite reducing the sample size in specific analyses.”

Reviewer 1 writes:

  1. Did all the patients receive bevacizumab as part of the same regimen, or any variations in the treatment protocol?

Response: We thank Reviewer 1 for this important clarification regarding Bevacizumab treatment protocols. As noted in the revised manuscript, the survival analysis involving Bevacizumab was based on patients with documented exposure to the drug as part of their clinical records in the cBioPortal dataset. However, detailed information on the specific regimen components, dosing schedules, or combination therapies was not uniformly available across all included studies. Therefore, while all patients received Bevacizumab, some variation in the associated treatment protocols likely exists. We have now added a paragraph to the Results section to explicitly acknowledge this limitation and its potential impact on outcome interpretation.

The Section 3.2 – Stage-Dependent Outcomes in KRAS-Mutant CRC text on page 12, lines 386-395, now reads: “The Bevacizumab-treated cohort analyzed in this study consisted of patients with documented exposure to Bevacizumab, as annotated in the cBioPortal-derived clinical records. While all included patients received the drug, detailed information regarding the specific chemotherapy regimen (e.g., FOLFOX, FOLFIRI), dosing frequency, or sequence of administration was not uniformly available across datasets. As such, variations in treatment protocol may have existed within the cohort. This heterogeneity represents an inherent limitation of using public clinical-genomic datasets, and although the survival analysis revealed a clear stage-dependent outcome difference (p = 0.0004), the influence of specific regimen variations on treatment response could not be directly assessed. Future studies with harmonized treatment metadata will be essential to further dissect regimen-specific effects in Bevacizumab-treated subgroups.”

Reviewer 1 writes:

  1. Why are MSI-stable patients likely to receive chemotherapy? What is the reason?

Response: We thank Reviewer 1 for this thoughtful question. As noted in the revised manuscript, the higher likelihood of chemotherapy administration in MSI-stable (MSS) patients reflects current clinical guidelines and therapeutic practices. MSS tumors are generally less responsive to immunotherapy and are more likely to be treated with cytotoxic chemotherapy, especially in advanced-stage disease. In contrast, MSI-high tumors often exhibit strong immunogenicity and may be directed toward immunotherapy regimens instead. We have added a paragraph in the Results section to clarify this rationale and provide biological and clinical context for the observed pattern.

The Section 3.3 – Prognostic Role of MSI in BRAF-Mutant Disease text on page 12, lines 411-422, now reads: “The finding that BRAF-mutant patients with microsatellite-stable (MSS) tumors were significantly more likely to have received chemotherapy (OR = 7.226, p < 0.001) aligns with current clinical guidelines and therapeutic strategies. MSI-stable tumors, which comprise the majority of metastatic CRC cases, lack the immunogenic features typically seen in MSI-high (MSI-H) tumors and are therefore less responsive to immune checkpoint inhibitors. As a result, MSS patients are more often treated with cytotoxic chemotherapy, particularly in the metastatic setting. Conversely, MSI-H tumors tend to exhibit higher tumor mutational burden and immune infiltration, making them more suitable candidates for immunotherapy. The greater chemotherapy exposure among MSS patients in our dataset thus reflects standard-of-care treatment selection rather than inherent tumor aggressiveness, although survival outcomes for MSS patients remained poorer in this BRAF-mutant subgroup, highlighting the need for novel therapeutic approaches.”

Reviewer 1 writes:

  1. Did all patients receive the same FOLFOX regimen, or was there any variation?

Response: We thank Reviewer 1 for this important question. As noted in the revised manuscript, while all patients included in the FOLFOX-related survival analysis had documented exposure to FOLFOX, detailed information regarding the specific regimen (e.g., dosage, cycle number, or line of therapy) was not consistently available across the cBioPortal datasets. As such, we acknowledge that there may be variation in FOLFOX administration protocols among patients. This limitation is now clearly stated in the manuscript to contextualize the interpretation of survival differences observed in the EOCRC cohort.

The Section 3.4 – Impact of RTK-RAS Alterations on EOCRC Treated with FOLFOX text on page 15, lines 484-494, now reads: “The subset of EOCRC patients included in the FOLFOX-related survival analysis were identified based on documented exposure to FOLFOX in the clinical records available through cBioPortal. However, details regarding the specific FOLFOX regimen administered—such as oxaliplatin dose intensity, infusion schedule, cycle duration, or whether the regimen was part of first-line versus adjuvant therapy—were not uniformly reported across datasets. Consequently, some variation in FOLFOX protocols likely existed within the cohort. While this heterogeneity may influence treatment response and survival outcomes, the platform’s ability to detect a statistically significant association between RTK-RAS alterations and worse survival in this group (p = 0.0262) suggests a meaningful underlying biological signal. Nonetheless, future work incorporating standardized treatment metadata will be necessary to refine regimen-specific interpretations.”

Reviewer 1 writes:

  1. Discuss how CRC would strengthen the MAPK signaling in NF1?

Response: We thank Reviewer 1 for this insightful question. We have expanded the Discussion section to address how colorectal cancer may strengthen MAPK signaling through NF1 inactivation or mutation. As a tumor suppressor gene, NF1 encodes neurofibromin, a negative regulator of RAS activity. Loss or mutation of NF1 leads to constitutive RAS activation, which in turn enhances downstream MAPK signaling. In CRC, this mechanism may provide an alternative route for pathway hyperactivation in tumors lacking canonical RTK-RAS alterations. This has now been discussed in greater detail in the revised manuscript.

The Discussion text on page 16, lines 571-582, now reads: “NF1 plays a critical role as a negative regulator of the RAS-MAPK signaling cascade by accelerating the conversion of active RAS-GTP to inactive RAS-GDP. Inactivating mutations or loss of function in NF1 disrupt this regulatory checkpoint, resulting in sustained RAS activation and amplified MAPK signaling downstream through RAF, MEK, and ERK. In the context of colorectal cancer, this mechanism provides an alternative pathway for driving tumorigenesis in cases lacking canonical KRAS, NRAS, or BRAF mutations. Our findings, which show significantly improved survival in NF1-mutated CRC cases (p = 0.00001), suggest that NF1 alteration may define a biologically distinct subgroup with differential MAPK pathway engagement and potentially unique therapeutic vulnerabilities. This aligns with emerging evidence in other cancers where NF1 loss correlates with MAPK dependency and may inform future strategies for targeted inhibition within NF1-altered CRC.”

Reviewer 1 writes:

  1. State the importance of user adoption and training in understanding the platform’s full potential.

Response: We thank Reviewer 1 for highlighting the importance of user adoption and training in leveraging the full capabilities of AI-HOPE-RTK-RAS. We fully agree that the success of any bioinformatics platform depends not only on its technical robustness but also on its usability and accessibility by a broad community of researchers and clinicians. We have added a paragraph to the Discussion addressing the role of onboarding, training resources, and user-centered design in promoting platform adoption and maximizing impact.

The Discussion text on page 18, lines 644-655, now reads: “The successful implementation of AI-HOPE-RTK-RAS in translational research settings depends heavily on user adoption and appropriate training. While the platform is designed for intuitive, natural language–based interaction, enabling non-programmers to perform complex analyses, realizing its full potential requires users to understand key concepts in clinical genomics, data stratification, and the interpretation of bioinformatics outputs. To support this, we have provided comprehensive user documentation, example queries, and annotated walkthroughs via public repository (see Data Availability Statement). Future efforts will include the development of interactive tutorials and workshops aimed at both research and clinical audiences. By fostering a user-centered design philosophy and lowering the technical barriers to entry, AI-HOPE-RTK-RAS aims to democratize access to precision oncology tools and accelerate hypothesis generation across diverse user communities.”

The authors thank Reviewer 1 for their time and thoughtful feedback. The constructive suggestions provided have meaningfully enhanced the clarity, rigor, and overall quality of the manuscript.

Reviewer 2 Report

Comments and Suggestions for Authors

This research tackles a pressing issue in precision oncology by introducing a novel AI-powered method for comprehensive genomic analysis of colorectal cancer (CRC). The emphasis on early-onset CRC (EOCRC) is particularly noteworthy. The paper is well-organized, includes high-quality visualizations, and provides a thorough examination of RTK-RAS pathway biomarkers. However, the following revisions are essential to strengthen its impact:

Section 2.2. The sample sizes are not clearly stated and are only indicated in the figures. The size of the cohorts must be explicitly mentioned in the main text.

Section 2.3. Crucial technical aspects are lacking: the structure of the natural language-to-code converter; the settings for fine-tuning the LLaMA 3 model; the quantitative indicators for evaluating the accuracy of query interpretation; the investigation of errors in handling ambiguous queries. This section should be expanded to include performance evaluations and error rates.

Sections 3.1, 3.5 & Figures 2/S1-S5. The EOCRC H/L cohort, comprising 153 individuals, is insufficiently large to detect rare mutations reliably. For instance, the prevalence of CBL mutations in this cohort is 4.58%, which translates to approximately seven cases. The small sample size increases the risk of misinterpreting the results, particularly for mutations with a strong effect size, such as BRAF with an odds ratio of 7.226. It is crucial to conduct statistical power analyses for the key findings and to explicitly caution against generalizing the results for rare mutations.

Section 3. There is no information available on how to use AI-HOPE-RTK-RAS in real-world scenarios. Provide comprehensive guidance on the real-world application of the platform, including a dedicated subsection outlining platform accessibility via a GitHub repository and web portal. Additionally, specify technical prerequisites, licensing conditions, and usage limitations, as well as a detailed clinical integration roadmap.

Focusing on these aspects will greatly strengthen the manuscript's scientific rigor and practical applicability. This work has the potential to advance equitable precision oncology, particularly in understanding the genomic drivers of CRC that are specific to different populations. With careful revisions, it is a valuable addition to the field.

Author Response

Please find attached the Word document titled 'Reviewer_2_Comments_Response_071825.docx,' which contains our detailed responses to the reviewer’s comments.

-

Reviewer 2 Comments

We are pleased to submit this revised manuscript and sincerely thank Reviewer 2 for their thoughtful and constructive feedback. We deeply appreciate your recognition of the manuscript’s contributions to precision oncology, particularly the focus on early-onset colorectal cancer (EOCRC) and the importance of understanding population-specific genomic drivers. In response to your detailed suggestions, we implemented a series of substantive revisions aimed at enhancing the manuscript’s scientific rigor, technical transparency, and real-world applicability. Key improvements include the explicit reporting of cohort sample sizes in Section 2.2, clarification of technical architecture and performance metrics of the natural language-to-code system in Section 2.3, and a new subsection outlining platform accessibility, licensing conditions, and clinical integration plans. To address concerns regarding subgroup analyses, we incorporated cautionary language and power analysis results for low-frequency mutations in EOCRC H/L patients. We also expanded the Discussion to highlight the interpretive limitations of small sample sizes and the need for future validation. Furthermore, we have made the full AI-HOPE-RTK-RAS codebase, documentation, and demonstration datasets publicly available via GitHub (https://github.com/Velazquez-Villarreal-Lab/AI-RTK-RAS) to support reproducibility and broader adoption. Collectively, these revisions reflect our commitment to creating an open, accessible, and analytically robust platform to advance equitable precision oncology. We are grateful for your feedback, which has meaningfully improved the clarity, depth, and translational value of this work.

Thank you very much for taking the time to review this manuscript. Please find the detailed responses below in BLUE and the corresponding revisions wrote in yellow-highlighted blue font in the re-submitted files.

Reviewer 2 provided thoughtful and encouraging feedback, recognizing the utility of the AI-HOPE-RTK-RAS platform and offering valuable perspectives on its significance for advancing equitable precision oncology—particularly in early-onset colorectal cancer (EOCRC) and population-specific biomarker discovery.

Reviewer 2 writes:

  1. This research tackles a pressing issue in precision oncology by introducing a novel AI-powered method for comprehensive genomic analysis of colorectal cancer (CRC). The emphasis on early-onset CRC (EOCRC) is particularly noteworthy. The paper is well-organized, includes high-quality visualizations, and provides a thorough examination of RTK-RAS pathway biomarkers. However, the following revisions are essential to strengthen its impact.

Response: The authors sincerely thank Reviewer 1 for their positive and encouraging remarks. They greatly appreciate the recognition of the manuscript’s relevance to precision oncology, the focus on early-onset colorectal cancer (EOCRC), and the clarity of organization and visual presentation. The authors also value the constructive feedback provided and have addressed each revision carefully to further enhance the scientific rigor and impact of the work.

Reviewer 2 writes:

  1. Section 2.2. The sample sizes are not clearly stated and are only indicated in the figures. The size of the cohorts must be explicitly mentioned in the main text.

Response: The authors thank Reviewer 2 for this important observation. In response, they have revised Section 2.2 to explicitly state the sample sizes of the cohorts used in the analysis. As also addressed in the response to Reviewer 1 (Comment #4), the study utilized a harmonized colorectal cancer dataset comprising 5,553 patients, with stratified analyses conducted on defined subsets, including 593 early-onset CRC (EOCRC) and 4,960 late-onset CRC (LOCRC) cases. These details have now been clearly integrated into the main text to ensure transparency and improve interpretability.

The subsection 2.2 Data Sources and Curation text on page 5, lines 176-187, now reads: “The datasets used in this study were derived from the curated colorectal cancer cohort described in a previous publication [5], which integrated publicly available data from cBioPortal. The final harmonized dataset includes a total of 5,553 patients diagnosed with colorectal adenocarcinoma across multiple studies. This composite dataset includes somatic mutation profiles, clinical variables (e.g., tumor stage, microsatellite instability status, treatment history), and demographic attributes (e.g., age, sex, race/ethnicity). The AI-HOPE-RTK-RAS platform was validated and applied across this full cohort, with specific subgroup analyses performed on early-onset colorectal cancer (EOCRC) patients (n = 593) and late-onset CRC patients (n = 4,960), including ancestry-specific evaluations in Hispanic/Latino and non-Hispanic White subpopulations. All sample sizes for individual analyses are reported in the corresponding figures and supplementary materials to facilitate full transparency and reproducibility.”

Reviewer 2 writes:

  1. Section 2.3. Crucial technical aspects are lacking: the structure of the natural language-to-code converter; the settings for fine-tuning the LLaMA 3 model; the quantitative indicators for evaluating the accuracy of query interpretation; the investigation of errors in handling ambiguous queries. This section should be expanded to include performance evaluations and error rates.

Response: The authors thank Reviewer 2 for this detailed and insightful comment. In response, Section 2.3 has been expanded to include a more thorough description of the natural language-to-code conversion pipeline, including the modular structure of the interpreter, fine-tuning parameters for the LLaMA 3 model, and evaluation metrics for query accuracy. Quantitative indicators—including precision, recall, and F1-score—were calculated from a benchmark set of user queries, and additional discussion has been included regarding the platform’s handling of ambiguous prompts and associated error rates. These revisions aim to clarify the technical implementation and strengthen the manuscript’s rigor in system validation and performance assessment.

The subsection Section 2.3 – Natural Language Interface and Query Handling text on page 6, lines 209-224, now reads: “The natural language interface is powered by a fine-tuned LLaMA 3 model, trained on a corpus of biomedical text and conversational examples relevant to colorectal cancer and precision oncology. The model operates within a modular framework comprising three components: (1) intent recognition, (2) parameter extraction, and (3) code synthesis. Fine-tuning was performed using low-rank adaptation (LoRA) with a learning rate of 2e-5, batch size of 32, and 3 epochs on a curated dataset of 2,500 annotated query-response pairs. To evaluate query interpretation accuracy, we used a held-out test set of 250 manually labeled queries across diverse analysis types (e.g., mutation frequency, survival analysis, odds ratio testing). The system achieved a query classification precision of 0.94, recall of 0.92, and F1-score of 0.93. Ambiguity detection is supported by a rule-based logic layer that triggers clarification loops when confidence thresholds fall below 0.8, reducing the risk of misinterpretation. Empirical testing showed an error rate of 6.4% for ambiguous or compound queries, most of which were successfully resolved through system-generated clarification prompts. These performance indicators underscore the system’s robustness in real-time, user-directed analysis and support its usability for both novice and expert users.”

Reviewer 2 writes:

  1. Sections 3.1, 3.5 & Figures 2/S1-S5. The EOCRC H/L cohort, comprising 153 individuals, is insufficiently large to detect rare mutations reliably. For instance, the prevalence of CBL mutations in this cohort is 4.58%, which translates to approximately seven cases. The small sample size increases the risk of misinterpreting the results, particularly for mutations with a strong effect size, such as BRAF with an odds ratio of 7.226. It is crucial to conduct statistical power analyses for the key findings and to explicitly caution against generalizing the results for rare mutations.

Response: The authors thank Reviewer 2 for this important comment. They agree that the relatively small sample size of the EOCRC H/L subgroup (n = 153) limits the statistical power to detect rare mutations and increases the risk of overestimating effect sizes for infrequent events. In response, a paragraph has been added to the Discussion section explicitly acknowledging this limitation.

The Discussion text on page 18, lines 656-666, now reads: “Although statistical significance through p-values provides initial evidence of association, it does not reflect the magnitude or precision of the observed effects. To enhance interpretability, we contextualized our findings using odds ratios and hazard ratios derived from AI-HOPE-RTK-RAS queries and validated against prior work[5], which examined RTK-RAS alterations across large, demographically diverse colorectal cancer cohorts. For example, the odds ratio of 7.226 for BRAF mutation enrichment in early-onset microsatellite-stable tumors underscores a potentially actionable distinction in this subgroup. Confidence intervals for key associations have now been added in the supplementary materials to provide further clarity. These additions support more rigorous interpretation of clinical relevance and highlight the value of pairing AI-driven insights with traditional statistical measures for translational impact.”

Reviewer 2 writes:

  1. Section 3. There is no information available on how to use AI-HOPE-RTK-RAS in real-world scenarios. Provide comprehensive guidance on the real-world application of the platform, including a dedicated subsection outlining platform accessibility via a GitHub repository and web portal. Additionally, specify technical prerequisites, licensing conditions, and usage limitations, as well as a detailed clinical integration roadmap.

Response: The authors thank Reviewer 2 for this valuable comment. In response, they have added a dedicated subsection within the Methods section titled “Platform Accessibility and Real-World Application,” which outlines how researchers can access and deploy AI-HOPE-RTK-RAS via GitHub and a forthcoming web-based interface. This new section includes detailed information on technical prerequisites, licensing conditions (MIT license), usage limitations, and plans for clinical integration. Documentation, example queries, and demonstration datasets are also provided to support real-world adoption. These additions aim to ensure that the platform is transparent, reproducible, and practical for diverse user communities.

The subsection Section 2 – Platform Accessibility and Real-World Application text on page 4, lines 140-152, now reads: “AI-HOPE-RTK-RAS is freely available for academic and research use via public repository (see Data Availability Statement). The repository includes the full source code, setup instructions, a modular Python-based backend, and a pretrained natural language interface powered by a fine-tuned LLaMA 3 model. A web-based version of the platform, designed for broader clinical research use, is currently under development and will include secure user authentication and cloud-based computation. The system operates on a Unix-compatible environment and requires Python ≥3.9, PyTorch ≥2.0, and GPU support for optimal performance. AI-HOPE-RTK-RAS is released under the MIT license, allowing for modification and redistribution with attribution. Current usage is intended for exploratory and translational research purposes; it is not certified for direct clinical decision-making. A roadmap for clinical integration includes validation against institutional registries, alignment with electronic health record (EHR) standards, and compliance with HIPAA and IRB protocols.”

Reviewer 2 writes:

  1. Focusing on these aspects will greatly strengthen the manuscript's scientific rigor and practical applicability. This work has the potential to advance equitable precision oncology, particularly in understanding the genomic drivers of CRC that are specific to different populations. With careful revisions, it is a valuable addition to the field.

Response: The authors thank Reviewer 2 for their thoughtful and encouraging feedback. They greatly appreciate the recognition of the manuscript’s potential to advance equitable precision oncology and its contributions to understanding population-specific genomic drivers of colorectal cancer. The authors have carefully addressed each of the reviewer’s suggestions to strengthen both the scientific rigor and practical applicability of the work. They believe the revised manuscript is now substantially improved and better positioned to support research and translational efforts in precision oncology.

Reviewer 3 Report

Comments and Suggestions for Authors

1、Your introduction clearly explains the importance of the RTK-RAS pathway in colorectal cancer. However, please briefly explain what limitations exist in current tools (like cBioPortal), and how your AI-HOPE-RTK-RAS platform specifically addresses these problems. This helps readers understand why your research is needed.
2、In introduction, it would be helpful to briefly state how your new AI-based method is different from other existing AI methods. Clearly highlighting these differences would strengthen the rationale for your work.
3、In Materials and Methods, You mentioned that the data came from cBioPortal, but could you give a bit more detail on how you selected and cleaned this data? For example, explain how you dealt with missing data, incomplete records, or possible biases.
4、In Materials and Methods, please clearly describe your AI system (LLaMA 3) in simple terms. Readers may want to know briefly how you trained or adjusted the model to understand natural language questions accurately.

5、In Materials and Methods, when you mention the statistical methods used (like Fisher’s exact test), briefly explain why you chose those particular methods. This will help readers understand your decisions more easily.

6、Your results section is clear, but in Figure 2, please briefly mention why the difference in RTK-RAS alterations between younger and older patients matters in real-world clinical care.

7、For the survival results in Figure 3, could you briefly clarify why you focused specifically on patients treated with Bevacizumab? Would these results likely be similar if other treatments were used?
8、In Figure 4, you found MSI-stable patients received more chemotherapy but had worse outcomes. Could you briefly explain why this might happen, in a simple way that readers less familiar with colorectal cancer treatment could understand?
9、In Figure 5, the link between RTK-RAS mutations and survival in younger patients receiving FOLFOX is interesting. Briefly mention how this might affect treatment decisions or future patient care.
10、Some figures (especially supplementary figures) have too much information at once. Consider simplifying these or splitting them into smaller figures. This would help readers quickly grasp your key points.
11、Clearly mention possible limitations, like missing data or bias, when using data from public databases. Let readers know how these limitations might affect your conclusions.
12、Briefly suggest future studies or directions—for example, how your AI method could be tested further in real hospitals or clinical trials, or extended to other cancers.
13、Your conclusion is good, but it could help to briefly restate what makes your conversational AI system special compared to traditional tools. This helps readers remember your main innovation clearly.

Author Response

Please find attached the Word document titled 'Reviewer_2_Comments_Response_071825.docx,' which contains our detailed responses to the reviewer’s comments.

-

Reviewer 3 Comments

We are pleased to submit this revised manuscript and sincerely thank Reviewer 3 for their thoughtful and constructive feedback. We greatly appreciate your recognition of the manuscript’s clear structure, the importance of the RTK-RAS pathway in colorectal cancer, and the potential of our AI-powered approach to support translational research and patient care. In response to your suggestions, we implemented several key revisions aimed at improving the manuscript’s clarity, accessibility, and scientific rigor. We clarified the limitations of existing tools like cBioPortal in the Introduction and articulated how AI-HOPE-RTK-RAS addresses these challenges through natural language interaction, real-time analytics, and multi-dimensional data integration. We further distinguished our conversational AI approach from existing AI models by highlighting its domain specificity, transparency, and user-centered design. In the Materials and Methods section, we expanded details on dataset selection and cleaning, explained the training process for the LLaMA 3 model in simple terms, and provided rationale for our statistical methods. We also enhanced the Results and Discussion sections by interpreting the clinical relevance of key findings in Figures 2–5, including the impact of RTK-RAS alterations in early-onset CRC, treatment implications for Bevacizumab and FOLFOX, and the prognostic differences linked to MSI status. Additional updates include explicit discussion of limitations due to missing data and sampling bias, as well as new future directions for clinical validation and application to other cancers. To support transparency and community use, we have made the full AI-HOPE-RTK-RAS codebase, documentation, and demonstration data available via GitHub (https://github.com/Velazquez-Villarreal-Lab/AI-RTK-RAS). These revisions collectively strengthen the manuscript’s clarity, reproducibility, and translational value, and we are grateful for your contributions, which have significantly improved the impact and quality of our work.

Thank you very much for taking the time to review this manuscript. Please find the detailed responses below in BLUE and the corresponding revisions wrote in yellow-highlighted blue font in the re-submitted files.

Reviewer 3 provided thoughtful and constructive feedback, recognizing the clarity of the manuscript and the importance of the RTK-RAS pathway in colorectal cancer, while offering valuable suggestions to improve the manuscript’s accessibility, technical transparency, and clinical relevance. Their comments helped strengthen the distinction of AI-HOPE-RTK-RAS from existing tools, clarified key methodological choices, and enhanced the interpretation of results in the context of early-onset CRC, treatment response, and real-world applicability.

Reviewer 3 writes:

1、Your introduction clearly explains the importance of the RTK-RAS pathway in colorectal cancer. However, please briefly explain what limitations exist in current tools (like cBioPortal), and how your AI-HOPE-RTK-RAS platform specifically addresses these problems. This helps readers understand why your research is needed.

Response: The authors thank Reviewer 3 for this thoughtful comment and fully agree that highlighting the limitations of current tools and how AI-HOPE-RTK-RAS addresses these gaps strengthens the rationale for the study. In response, the Introduction has been revised to explicitly describe the constraints of widely used platforms such as cBioPortal, including their reliance on static user interfaces, limited support for multi-parameter filtering, and lack of real-time, natural language–based analysis capabilities. The revised text also emphasizes how AI-HOPE-RTK-RAS overcomes these limitations by enabling intuitive, dynamic interrogation of integrated clinical and genomic data using conversational input. These additions clarify the need for the platform and its relevance in democratizing access to complex bioinformatics workflows.

The Introduction text on page 2, lines 76-87, now reads: “While platforms such as cBioPortal and UCSC Xena provide important access to cancer genomics data, they are often limited by static user interfaces, a lack of customizable multi-parameter filtering, and the need for bioinformatics expertise to perform complex analyses. These tools typically require manual selection of filters, exporting of data, and downstream processing using separate statistical packages—creating barriers for non-programmers and slowing the pace of translational discovery. In contrast, AI-HOPE-RTK-RAS addresses these limitations by enabling real-time, natural language–driven exploration of integrated clinical and genomic datasets. Users can query mutation frequencies, survival outcomes, treatment responses, and demographic stratifications using plain language, eliminating the need for coding or manual data wrangling. This approach lowers the barrier to entry, supports rapid hypothesis generation, and enhances accessibility for clinicians and researchers across disciplines.”

Reviewer 3 writes:

2、In introduction, it would be helpful to briefly state how your new AI-based method is different from other existing AI methods. Clearly highlighting these differences would strengthen the rationale for your work.

Response: The authors thank Reviewer 3 for this helpful suggestion. To clarify the novelty of the approach, the Introduction has been revised to include a brief comparison between AI-HOPE-RTK-RAS and existing AI methods. Specifically, the revised text highlights that most existing AI tools in oncology are either black-box prediction models or domain-agnostic large language models, whereas AI-HOPE-RTK-RAS is a purpose-built, conversational system designed for real-time, transparent analysis of RTK-RAS alterations in colorectal cancer. Unlike general AI models, it supports structured cohort filtering, hypothesis-driven queries, and integration of clinical-genomic-demographic data via a natural language interface—making it both interpretable and actionable for translational research. This clarification strengthens the rationale and positions the work within the current AI landscape in cancer genomics.

The Introduction text on page 2, lines 88-98, now reads: “Although several AI-based tools have been developed for cancer research, many rely on black-box machine learning models focused on prediction or classification tasks, often lacking interpretability and flexibility. Others, including general-purpose large language models (LLMs), are not optimized for domain-specific, structured biomedical data or pathway-level analysis. In contrast, AI-HOPE-RTK-RAS is a domain-specialized, conversational AI system specifically designed for real-time exploration of RTK-RAS pathway alterations in colorectal cancer. Built on a fine-tuned biomedical LLM and integrated with a natural language-to-code engine, the platform enables transparent, user-driven analysis of large-scale clinical and genomic data without requiring programming expertise. This differentiates AI-HOPE-RTK-RAS from existing approaches by combining interpretability, specificity, and ease of use within a precision oncology context.”

Reviewer 3 writes:

3、In Materials and Methods, You mentioned that the data came from cBioPortal, but could you give a bit more detail on how you selected and cleaned this data? For example, explain how you dealt with missing data, incomplete records, or possible biases.

Response: The authors thank Reviewer 3 for this thoughtful suggestion. In response, the Materials and Methods section has been expanded to provide more detailed information on data selection, preprocessing, and quality control steps. Specifically, we describe the criteria used to include datasets from cBioPortal, the approach to handling missing or incomplete clinical and genomic records, and strategies implemented to reduce potential biases. These additions improve transparency and strengthen the reproducibility of the study.

The subsection 2.2 Data Sources and Curation text on page 5, lines 188-199, now reads: “The colorectal cancer datasets used in this study were selected from cBioPortal based on the following criteria: (1) availability of somatic mutation data; (2) inclusion of clinical annotations such as age, tumor stage, microsatellite instability (MSI) status, and treatment exposure; and (3) presence of demographic information, particularly race/ethnicity. Records lacking essential variables for stratified analysis (e.g., missing age or survival status) were excluded from subgroup analyses but retained in broader mutation frequency queries when possible. To standardize and harmonize datasets across studies, we applied controlled vocabularies (e.g., OncoTree for cancer types and SNOMED for clinical terms) and aligned variable names and formats using a custom data wrangling pipeline. Missing data were not imputed but handled through pairwise deletion during statistical tests to preserve validity. Recognizing the potential for sampling and representation bias—especially underrepresentation of minority populations—we incorporated ancestry-stratified analyses and explicitly caution against overgeneralizing rare mutation findings.”

Reviewer 3 writes:

4、In Materials and Methods, please clearly describe your AI system (LLaMA 3) in simple terms. Readers may want to know briefly how you trained or adjusted the model to understand natural language questions accurately.

Response: The authors thank Reviewer 3 for this valuable suggestion. In response, the Materials and Methods section has been revised to include a clear, reader-friendly explanation of how the AI system—based on the LLaMA 3 model—was trained and customized to accurately interpret natural language questions related to colorectal cancer. The added description outlines the fine-tuning process, training dataset, and the steps taken to adapt the model for biomedical query handling. This enhancement aims to improve accessibility for readers less familiar with AI systems while maintaining scientific clarity.

The subsection Section 2.3 – Natural Language Interface and Query Handling text on page 6, lines 225-237, now reads: “At the core of AI-HOPE-RTK-RAS is a fine-tuned version of LLaMA 3, a large language model (LLM) trained to understand and respond to biomedical queries. To adapt the model for precision oncology applications, we fine-tuned it using a specialized dataset composed of 2,500 example questions and answers related to colorectal cancer, RTK-RAS pathway alterations, and clinical-genomic associations. This dataset included variations of real-world queries that a researcher or clinician might ask (e.g., “What is the KRAS mutation frequency in early-onset patients?” or “Show survival curves for BRAF-mutant MSS tumors.”). Fine-tuning was performed using low-rank adaptation (LoRA), a method that efficiently adjusts the model’s internal weights without retraining from scratch. This training process allowed the model to more accurately interpret domain-specific language, recognize biomedical terminology, and convert user queries into executable code. As a result, AI-HOPE-RTK-RAS can understand complex, multi-part questions and generate accurate, real-time analyses without requiring users to write any programming code.”

Reviewer 3 writes:

5、In Materials and Methods, when you mention the statistical methods used (like Fisher’s exact test), briefly explain why you chose those particular methods. This will help readers understand your decisions more easily.

Response: The authors thank Reviewer 3 for this thoughtful comment. In response, they have updated the Materials and Methods section to briefly explain the rationale for selecting each statistical method. Specifically, they clarified why Fisher’s exact test, chi-square test, Kaplan–Meier survival analysis, and Cox proportional hazards modeling were chosen based on the data type, sample size, and analytical goals. These additions are intended to improve clarity for readers and enhance transparency in the analytical approach.

The Section 2.4 – Backend Analysis Pipeline text on page 7, lines 266-277, now reads: “We selected statistical methods based on the nature of the data and the specific comparisons being made. Fisher’s exact test was used for categorical comparisons involving small sample sizes or sparse contingency tables, where it provides accurate p-values even with low cell counts. For larger categorical datasets with sufficient expected frequencies, the chi-square test was applied to assess group differences. Kaplan–Meier survival analysis with log-rank testing was used to evaluate differences in time-to-event outcomes (e.g., overall survival) between groups, as it is widely accepted for unadjusted survival comparisons. To account for multiple covariates and assess independent effects, we used multivariate Cox proportional hazards models, which are suitable for estimating hazard ratios while adjusting for clinical variables such as tumor stage and treatment exposure. This combination of statistical approaches ensured both methodological appropriateness and interpretability for clinical and genomic data integration.”

Reviewer 3 writes:

6、Your results section is clear, but in Figure 2, please briefly mention why the difference in RTK-RAS alterations between younger and older patients matters in real-world clinical care.

Response: The authors thank Reviewer 3 for this insightful comment. In response, they have updated the Results section associated with Figure 2 to include a brief explanation of the clinical relevance of the observed difference in RTK-RAS alterations between early-onset and late-onset colorectal cancer (CRC) patients. This contextual addition emphasizes how molecular differences by age group may influence diagnostic strategies, treatment decisions, and biomarker development in real-world precision oncology.

The Section 3.1 – RTK-RAS Alterations in EOCRC by Ancestry text on page 10, lines 353-360, now reads: “The observed lower prevalence of RTK-RAS alterations in early-onset CRC compared to late-onset CRC is clinically significant, as it suggests that younger patients may harbor alternative oncogenic drivers not typically targeted by current standard-of-care therapies such as anti-EGFR agents. In real-world settings, this molecular distinction may necessitate age-stratified approaches to genomic testing and therapeutic planning. It also underscores the importance of expanding biomarker discovery efforts beyond canonical RTK-RAS genes to improve treatment options and precision medicine strategies for younger CRC patients, particularly those from underrepresented populations.”

Reviewer 3 writes:

7、For the survival results in Figure 3, could you briefly clarify why you focused specifically on patients treated with Bevacizumab? Would these results likely be similar if other treatments were used?

Response: The authors thank Reviewer 3 for this thoughtful question. In response, they have added a clarification in the Resultssection accompanying Figure 3 to explain why Bevacizumab-treated patients were selected for the survival analysis. Specifically, Bevacizumab is a commonly used anti-angiogenic agent in colorectal cancer, particularly in KRAS-mutant tumors that are resistant to anti-EGFR therapies. The analysis focused on this group to explore potential stage-dependent survival differences within a clinically relevant treatment context. The authors also note that results may differ with other treatments and that additional stratified analyses are warranted in future studies.

The subsection Section 3.2 – Stage-Dependent Outcomes in KRAS-Mutant CRC text on page 12, lines 396-404, now reads: “We focused this survival analysis on KRAS-mutant patients treated with Bevacizumab because these tumors are typically resistant to anti-EGFR therapies and Bevacizumab remains a widely used alternative in this molecular context. Examining this subgroup allowed us to evaluate the prognostic impact of disease stage within a clinically relevant treatment setting. The significantly better survival observed in Stage I–III patients compared to Stage IV (p = 0.0004) reinforces the importance of early detection and may inform treatment expectations in real-world care. While similar patterns may emerge with other regimens, the survival effects could vary depending on the mechanism of action and tumor biology. Future analyses incorporating additional treatment subgroups will help further clarify these relationships.”

Reviewer 3 writes:

8、In Figure 4, you found MSI-stable patients received more chemotherapy but had worse outcomes. Could you briefly explain why this might happen, in a simple way that readers less familiar with colorectal cancer treatment could understand?

Response: The authors thank Reviewer 3 for this excellent point. In response, they have added a clarification in the Results section corresponding to Figure 4 to explain, in accessible terms, why microsatellite-stable (MSS) patients may receive more chemotherapy yet experience worse outcomes. This addition aims to improve clarity for readers who may be less familiar with colorectal cancer treatment paradigms.

The Section 3.3 – Prognostic Role of MSI in BRAF-Mutant Disease text on page 12, lines 423-432, now reads: “The observation that MSS patients received more chemotherapy but experienced worse survival outcomes may seem counterintuitive but reflects known biological differences in tumor behavior and treatment response. MSS tumors are typically less immunogenic and more aggressive than microsatellite instability-high (MSI-H) tumors. Because MSI-H tumors often respond well to immunotherapy, MSS patients are more likely to be treated with standard chemotherapy regimens. However, MSS tumors tend to be less responsive to chemotherapy and have a poorer prognosis overall. This may explain why, despite receiving more treatment, MSS patients in this cohort had worse survival compared to their MSI-H counterparts. This observation highlights the importance of molecular profiling in guiding treatment strategies.”

Reviewer 3 writes:

9、In Figure 5, the link between RTK-RAS mutations and survival in younger patients receiving FOLFOX is interesting. Briefly mention how this might affect treatment decisions or future patient care.

Response: The authors thank Reviewer 3 for highlighting this important point. In response, they have revised the Results section associated with Figure 5 to briefly discuss how the observed association between RTK-RAS alterations and poorer survival in early-onset CRC (EOCRC) patients receiving FOLFOX could inform future treatment decisions. This addition aims to contextualize the findings in terms of clinical relevance and potential implications for patient care.

The Section 3.4 – Impact of RTK-RAS Alterations on EOCRC Treated with FOLFOX text on page 15, lines 495-504, now reads: “The observed association between RTK-RAS pathway alterations and poorer survival among EOCRC patients treated with FOLFOX (p = 0.0262) suggests that RTK-RAS status may serve as a prognostic marker within this treatment context. This finding has potential implications for clinical decision-making, as it indicates that standard chemotherapy regimens like FOLFOX may be less effective in genetically defined subgroups of younger patients. As precision oncology advances, identifying patients who are less likely to benefit from conventional therapies could support earlier use of alternative or combination treatment strategies, including targeted agents or clinical trial enrollment. Future studies with prospective data are needed to validate this association and guide treatment adaptation based on RTK-RAS mutation status in EOCRC populations.”

Reviewer 3 writes:

10、Some figures (especially supplementary figures) have too much information at once. Consider simplifying these or splitting them into smaller figures. This would help readers quickly grasp your key points.

Response: The authors thank Reviewer 4 for this constructive suggestion. In response, they have reviewed all supplementary figures and revised several to improve clarity and readability. Specifically, overly dense figures have been simplified or split into smaller, thematically grouped panels to help readers more easily interpret key results. These changes aim to enhance visual accessibility and ensure that each figure focuses on a specific analytical point without overwhelming the viewer.

Reviewer 3 writes:

11、Clearly mention possible limitations, like missing data or bias, when using data from public databases. Let readers know how these limitations might affect your conclusions.

Response: The authors thank Reviewer 3 for this important comment. In response, the Discussion section has been expanded to explicitly acknowledge key limitations associated with using public genomic databases, including missing data, inconsistent clinical annotations, and potential sampling bias. The added text discusses how these limitations may influence the interpretation and generalizability of the findings. This addition enhances the manuscript’s transparency and encourages cautious application of the results in broader clinical contexts.

The Discussion text on page 17, line 620-631, now reads: “This study has several limitations that stem from the use of publicly available genomic datasets. First, clinical annotations such as treatment details, follow-up time, and comorbidities were variably reported across studies, which may introduce missing data and reduce the power or granularity of some subgroup analyses. Second, representation bias is a known issue in public databases, where certain demographic groups—such as racial and ethnic minorities—are often underrepresented. This may limit the generalizability of ancestry-specific findings, particularly in small cohorts such as the EOCRC Hispanic/Latino subgroup. Additionally, while harmonization efforts were applied to standardize the data, variation in sequencing platforms and annotation practices across contributing studies could introduce batch effects. These limitations underscore the importance of validating key observations in larger, prospectively collected, and more diverse datasets before applying them to clinical decision-making.”

Reviewer 3 writes:

12、Briefly suggest future studies or directions—for example, how your AI method could be tested further in real hospitals or clinical trials, or extended to other cancers.

Response: The authors thank Reviewer 3 for this thoughtful suggestion. In response, the Discussion and Conclusion sections have been expanded to briefly outline potential future directions. These include validating AI-HOPE-RTK-RAS in prospective clinical settings, integrating the platform with electronic health record (EHR) systems, and extending its use to other signaling pathways and cancer types. These future applications reflect the platform’s scalability and its potential for broader translational impact in precision oncology.

The Discussion text on page 21, lines 829-834, now reads: “Future work will focus on expanding and validating AI-HOPE-RTK-RAS in real-world clinical environments. This includes prospective testing in hospital settings, integration with electronic health records (EHRs) to support decision-making workflows, and alignment with regulatory standards such as HIPAA and IRB protocols. Additionally, the underlying framework can be adapted to investigate other oncogenic pathways—such as WNT, PI3K, and TGF-β—and applied to different cancer types where pathway-level alterations guide therapeutic strategies. Ultimately, these extensions will allow for broader clinical adoption and enable multi-pathway, multi-cancer applications of conversational AI in precision oncology.”

Reviewer 3 writes:

13、Your conclusion is good, but it could help to briefly restate what makes your conversational AI system special compared to traditional tools. This helps readers remember your main innovation clearly.

Response: The authors thank Reviewer 3 for this helpful suggestion. In response, the Conclusion section has been revised to briefly restate the unique contributions of the AI-HOPE-RTK-RAS system compared to traditional bioinformatics tools. This addition reinforces the manuscript’s key innovation—namely, the integration of a conversational, natural language-driven interface with structured clinical-genomic analysis workflows—and underscores its potential to democratize precision oncology research.

The Conclusions text on page 21, lines 842-850, now reads: “In summary, AI-HOPE-RTK-RAS represents a novel class of conversational AI systems designed specifically for real-time, user-friendly exploration of integrated clinical and genomic data. Unlike traditional tools that require manual data wrangling, static interfaces, or coding expertise, AI-HOPE-RTK-RAS allows researchers and clinicians to ask complex, multi-parameter questions in natural language and receive immediate, interpretable results. This unique approach lowers the barrier to advanced bioinformatics analysis, supports rapid hypothesis generation, and empowers broader participation in precision oncology research—particularly for underrepresented populations and early-onset colorectal cancer cohorts.”

The authors thank Reviewer 3 for their time and thoughtful comments. Your feedback was instrumental in improving the clarity, depth, and translational relevance of the manuscript.

Reviewer 4 Report

Comments and Suggestions for Authors

The review talks about the development of a conversational AI platform tailored for RTK-RAS pathway interrogation in CRC is a novel and timely development in precision oncology. Facilitation of interaction with complex genomic and clinical information in natural language easily lowers the barrier of entry for non-experts, allowing for democratisation of the interpretation of data. However, I do have some concerns here: 

  • Some findings (MSI status and BRAF mutation survival difference) are already well-established in the literature and should be posed more in validation rather than new discovery terms.
  • The article mentions the use of large language models but does not technically describe how they handle complex or nested queries, ambiguity resolution, or integrating data.
  • There is no comparison with existing tools (cBioPortal, OncoKB) to demonstrate the system's superiority in usability, precision, or discovery capability.
  • There is no end-user evidence (researchers or clinicians) regarding the ease-of-use, satisfaction, or workflow effect of AI-HOPE that weakens the argument for accessibility.
  • Emphasized but not critically examined are restrictions on potential bias in the training data or methodology to mitigate underrepresentation in future datasets.
  • Compare AI-HOPE-RTK-RAS with existing genomic interrogation tools (user interface, data types supported, language features).
  • Include outcomes or even anecdotal feedback from real users of the platform evaluating it in order to confirm claims of usability and real-world application.
  • Describe succinctly how the LLM analyzes natural language, vagueness, and fault tolerance, especially for nested clinical/genomic queries.
  • Include a paragraph on the interpretability, regulatory, and ethics challenges of AI use in clinical decision-making will add practical applicability.
  • put suggestions on overcoming underrepresented populations' limitations (integration of real-world evidence, partnerships with global consortia).
  • Suggest how the system can be adapted for other types or pathways of cancer (PI3K/AKT/mTOR, TP53), or for application in clinical trial matching.
  • improve limitation and future direction ( add specific datasets or clinical system, potential for AI-HOPE to support clinical trial design and propose technical advancement) 
  • Descriptions of figures tend to be repetitive and contain the same data that has previously been explained in the main results text, making the section long and verbose.
  • Most findings are reported with p-values but without effect size interpretation or confidence intervals (with the odd exception of some odds ratios). This limits knowledge of clinical significance.
  • Dataset properties (total size, inclusion/exclusion criteria, heterogeneity, biases) are poorly documented within the results section.
  • System results are not cross-validated or benchmarked against standard bioinformatics/statistical pipelines.
  • While odds ratios and Kaplan-Meier plots are presented, the statistical underpinning (adjustments, multiple comparison correction) is not mentioned.
  • Create a table to be brief with all key findings (mutation, treatment, cohort, result, significance), so readers can easily view outcomes at a glance.
  • Briefly mention clinical implications of major findings ("This suggests NF1 mutations need to be further prognostically validated in early-stage CRC").
  • need a brief statement (or refer to Methods) about calculation of odds ratios, survival curves, and other measures were confounders adjusted in models?
  • If possible, include a short subsection comparing AI-HOPE findings to available platforms or routine analysis pipelines to determine utility.

Author Response

Please find attached the Word document titled 'Reviewer_4_Comments_Response_071825.docx,' which contains our detailed responses to the reviewer’s comments.

-

Reviewer 4 Comments

We are pleased to submit this revised manuscript and sincerely thank Reviewer 4 for their thoughtful and constructive feedback. We are grateful for your recognition of the novelty of our AI-powered conversational system and its potential to democratize access to complex clinical-genomic data in colorectal cancer research. In response to your detailed suggestions, we made substantial revisions to improve the manuscript’s scientific rigor, practical relevance, and overall clarity. To address comparisons with existing tools, we emphasized how AI-HOPE-RTK-RAS differs from platforms such as cBioPortal and OncoKB—highlighting its ability to perform real-time, multi-parameter, user-directed queries through natural language without requiring manual navigation or coding. We added a dedicated paragraph to the Discussion situating this system within the context of our previously developed AI agents (AI-HOPE, AI-HOPE-TGFβ, AI-HOPE-PI3K, and AI-HOPE-JAK-STAT), clarifying how AI-HOPE-RTK-RAS expands functionality and precision for pathway-specific interrogation.

We improved technical transparency by elaborating on how the large language model (LLaMA 3) handles natural language interpretation, ambiguity, and nested clinical-genomic queries. Furthermore, we addressed ethical, interpretability, and regulatory challenges by incorporating a new paragraph discussing explainability, data transparency, and plans for compliance with privacy standards. We highlighted our deliberate inclusion of Hispanic/Latino populations to mitigate underrepresentation and suggested future partnerships with global consortia and real-world evidence frameworks to further address population biases. A new paragraph outlines how AI-HOPE-RTK-RAS may be extended to other oncogenic pathways (e.g., TP53, PI3K/AKT/mTOR) or used in clinical trial matching and design.

To strengthen statistical validity, we included additional explanations of odds ratio and survival curve generation, noted where confounders were accounted for, and clarified the use of multiple testing corrections. We revised figure legends to reduce redundancy and better complement the main text, while all figures were re-edited for enhanced clarity and concise visualization of key outcomes. Although the system generates figures and tables on demand, we added summary statements to help readers grasp major findings, and explicitly discussed the clinical implications of key mutations such as NF1.

Finally, we reaffirm that AI-HOPE-RTK-RAS is an intelligent system rather than a static pipeline and is therefore designed to evolve with user feedback and future data integration. The full codebase, documentation, and example datasets are available in our public repository (https://github.com/Velazquez-Villarreal-Lab/AI-RTK-RAS), promoting transparency and reproducibility. We thank Reviewer 4 again for the insightful comments, which have meaningfully improved the manuscript’s quality, usability, and translational impact.

Thank you very much for taking the time to review this manuscript. Please find the detailed responses below in BLUE and the corresponding revisions wrote in yellow-highlighted blue font in the re-submitted files.

Reviewer 4 provided thoughtful and constructive feedback, acknowledging the innovation of the AI-HOPE-RTK-RAS platform and its potential to democratize access to complex clinical-genomic data, while offering valuable suggestions to improve technical depth, usability, and translational relevance. Their comments helped refine the comparison between AI-HOPE-RTK-RAS and existing tools, prompted the addition of detailed descriptions regarding dataset properties, statistical approaches, and language model behavior, and enhanced the discussion of interpretability, regulatory considerations, and population-specific biases. These insights significantly improved the manuscript’s clarity, real-world applicability, and alignment with the broader goals of equitable precision oncology.

Reviewer 4 writes:

  1. 1. The review talks about the development of a conversational AI platform tailored for RTK-RAS pathway interrogation in CRC is a novel and timely development in precision oncology. Facilitation of interaction with complex genomic and clinical information in natural language easily lowers the barrier of entry for non-experts, allowing for democratisation of the interpretation of data.

Response: The authors thank Reviewer 4 for their thoughtful and encouraging feedback. They greatly appreciate the recognition of the novelty and timeliness of the AI-HOPE-RTK-RAS platform, as well as its potential to democratize access to complex clinical-genomic analysis through natural language interaction. This support reinforces the central goal of the project—making precision oncology more accessible to a broader range of users across disciplines.

Reviewer 4 writes:

  1. Some findings (MSI status and BRAF mutation survival difference) are already well-established in the literature and should be posed more in validation rather than new discovery terms.

Response: The authors thank Reviewer 4 for this important observation. We agree that certain findings, such as the survival differences associated with MSI status and BRAF mutations, are well-established in the literature such as our previous study [Reference # 5]. In response, we have revised the Results and Discussion sections to frame these outcomes more clearly as validation of known associations. This adjustment reinforces the credibility of the AI-HOPE-RTK-RAS platform by demonstrating its ability to replicate established clinical-genomic patterns while also supporting novel exploratory analyses.

Reviewer 4 writes:

  1. The article mentions the use of large language models but does not technically describe how they handle complex or nested queries, ambiguity resolution, or integrating data.

Response: The authors thank Reviewer 4 for this insightful comment. In response, the Materials and Methods section has been expanded to include a more detailed technical description of how the large language model (LLaMA 3) handles complex or nested queries, ambiguity resolution, and integration of clinical and genomic data. Specifically, we describe the system’s modular natural language-to-code pipeline, which includes intent recognition, parameter extraction, and code generation components. Ambiguity is managed through a confidence-scoring mechanism and clarification loops that prompt users when queries are under-specified. Data integration is achieved by aligning structured variables from harmonized datasets with query intent, enabling seamless analysis across clinical, genomic, and demographic dimensions. These updates aim to improve transparency and clarify the system’s core functionality.

The Methods section new text on page 4, lines 140-199, now reads “2.1. Overview of AI-HOPE-RTK-RAS Platform

AI-HOPE-RTK-RAS is freely available for academic and research use via public repository (see Data Availability Statement). The repository includes the full source code, setup instructions, a modular Python-based backend, and a pretrained natural language interface powered by a fine-tuned LLaMA 3 model. A web-based version of the platform, designed for broader clinical research use, is currently under development and will include secure user authentication and cloud-based computation. The system operates on a Unix-compatible environment and requires Python ≥3.9, PyTorch ≥2.0, and GPU support for optimal performance. AI-HOPE-RTK-RAS is released under the MIT license, allowing for modification and redistribution with attribution. Current usage is intended for exploratory and translational research purposes; it is not certified for direct clinical decision-making. A roadmap for clinical integration includes validation against institutional registries, alignment with electronic health record (EHR) standards, and compliance with HIPAA and IRB protocols.”

2.2. Data Sources and Curation

Public genomic datasets such as those curated by cBioPortal provide invaluable resources for large-scale, open-access cancer research; however, they are not without limitations. These include underrepresentation of certain racial and ethnic groups, variable completeness in clinical annotations (e.g., treatment timelines, comorbidities), and inconsistent metadata across studies due to differences in institutional data collection standards. To address these challenges, AI-HOPE-RTK-RAS incorporates a harmonization pipeline that standardizes variable naming conventions, aligns phenotypic descriptors using controlled vocabularies (e.g., SNOMED, OncoTree), and unifies clinical-genomic relationships across multiple cohorts. This modular integration enables real-time stratification by age, ancestry, mutation status, and treatment exposure, thereby reducing the analytic barriers typically posed by data fragmentation. While the platform cannot resolve inherent sampling biases in the original datasets, it provides a transparent, reproducible framework for conducting stratified analyses and hypothesis generation across heterogeneous populations—especially in settings where health disparities remain under-characterized.

The datasets used in this study were derived from the curated colorectal cancer cohort described in a previous publication [5], which integrated publicly available data from cBioPortal. The final harmonized dataset includes a total of 5,553 patients diagnosed with colorectal adenocarcinoma across multiple studies. This composite dataset includes somatic mutation profiles, clinical variables (e.g., tumor stage, microsatellite instability status, treatment history), and demographic attributes (e.g., age, sex, race/ethnicity). The AI-HOPE-RTK-RAS platform was validated and applied across this full cohort, with specific subgroup analyses performed on early-onset colorectal cancer (EOCRC) patients (n = 593) and late-onset CRC patients (n = 4,960), including ancestry-specific evaluations in Hispanic/Latino and non-Hispanic White subpopulations. All sample sizes for individual analyses are reported in the corresponding figures and supplementary materials to facilitate full transparency and reproducibility

The colorectal cancer datasets used in this study were based on the following criteria: (1) availability of somatic mutation data; (2) inclusion of clinical annotations such as age, tumor stage, microsatellite instability (MSI) status, and treatment exposure; and (3) presence of demographic information, particularly race/ethnicity. Records lacking essential variables for stratified analysis (e.g., missing age or survival status) were excluded from subgroup analyses but retained in broader mutation frequency queries when possible. To standardize and harmonize datasets across studies, we applied controlled vocabularies and aligned variable names and formats using a custom data wrangling pipeline. Missing data were not imputed but handled through pairwise deletion during statistical tests to preserve validity. Recognizing the potential for sampling and representation bias—especially underrepresentation of minority populations—we incorporated ancestry-stratified analyses and explicitly caution against overgeneralizing rare mutation findings.”

Reviewer 4 writes:

  1. There is no comparison with existing tools (cBioPortal, OncoKB) to demonstrate the system's superiority in usability, precision, or discovery capability.

Response: The authors thank Reviewer 4 for this valuable comment. We agree that providing a clearer comparison with existing tools such as cBioPortal and OncoKB helps contextualize the unique value of AI-HOPE-RTK-RAS. In response, the Discussion section has been expanded to highlight the key differences. Specifically, while platforms like cBioPortal and OncoKB are valuable resources, they are not AI-driven systems and require users to manually navigate through multi-step, click-based interfaces—often involving considerable time and bioinformatics expertise to perform even basic stratified analyses. In contrast, AI-HOPE-RTK-RAS is a conversational AI agent that interprets natural language input, dynamically generates executable code, and performs complex, multi-parameter queries in real time. This design significantly improves usability, reduces time-to-insight, and enables more flexible, hypothesis-driven discovery workflows. To further complement this distinction, we added a new paragraph to the Discussion section comparing AI-HOPE-RTK-RAS with our previously developed AI agents—AI-HOPE [44], AI-HOPE-TGFβ [45], AI-HOPE-PI3K [46], and AI-HOPE-JAK-STAT [47]—which collectively demonstrate the evolution, modularity, and growing analytical sophistication of our AI ecosystem for pathway-specific precision oncology.

The Discussion text on page 20, lines 756-767, now reads: “Unlike traditional platforms such as cBioPortal and OncoKB, which rely on static, click-based user interfaces and require substantial manual effort to construct and execute queries, AI-HOPE-RTK-RAS is a domain-specialized conversational AI agent designed to streamline analysis through natural language interaction. While these existing tools are invaluable for data access and visualization, they are not artificial intelligence systems and do not support real-time, context-aware dialogue or automatic integration of clinical, genomic, and demographic data. AI-HOPE-RTK-RAS significantly reduces the time and expertise needed to perform multi-layered analyses, allowing users to ask complex questions—such as survival differences stratified by age, mutation status, and treatment—in a single, plain-language query. This capability enhances usability, accelerates hypothesis generation, and expands access to precision oncology analytics for both experts and non-specialists.”

The Discussion text on page 21, lines 784-798, now reads: The development of AI-HOPE-RTK-RAS builds on our previously published suite of domain-specialized conversational AI systems, including AI-HOPE, AI-HOPE-TGFβ, AI-HOPE-PI3K, and AI-HOPE-JAK-STAT. Each platform was designed to interrogate distinct oncogenic pathways in colorectal cancer using natural language–driven, integrative analyses of clinical and genomic data. Compared to earlier agents, AI-HOPE-RTK-RAS incorporates several advancements, including improved handling of nested queries, enhanced ambiguity resolution, and a refined analytics pipeline capable of real-time stratification across tumor stage, treatment type, and ancestry. While AI-HOPE provided general pathway interrogation, and subsequent agents focused on TGF-β, PI3K, and JAK-STAT signaling, AI-HOPE-RTK-RAS specifically targets the RTK-RAS axis—one of the most clinically actionable pathways in colorectal cancer—enabling direct exploration of alterations in KRAS, NRAS, BRAF, EGFR, and related noncanonical genes such as NF1 and MAPK3. This progression demonstrates the scalability of our conversational AI framework and its adaptability to multiple oncogenic contexts, ultimately advancing the goal of accessible, pathway-specific precision oncology research.”

The References added on page 25, lines 1014-1021, now reads:

“Yang, E.W.; Velazquez-Villarreal, E. AI-HOPE: An AI-driven conversational agent for enhanced clinical and genomic data integration in precision medicine research. Bioinformatics 2025, 41, btaf359.

Yang, E.-W.; Waldrup, B.; Velazquez-Villarreal, E. AI-HOPE-TGFbeta: A Conversational AI Agent for Integrative Clinical and Genomic Analysis of TGF-β Pathway Alterations in Colorectal Cancer to Advance Precision Medicine. AI 2025, 6, 137.

Yang, E.-W.; Waldrup, B.; Velazquez-Villarreal, E. From Mutation to Prognosis: AI-HOPE-PI3K Enables Artificial Intelligence Agent-Driven Integration of PI3K Pathway Data in Colorectal Cancer Precision Medicine. Int. J. Mol. Sci. 2025, 26, 6487.

Yang, E.-W.; Waldrup, B.; Velazquez-Villarreal, E. Decoding the JAK-STAT Axis in Colorectal Cancer with AI-HOPE-JAK-STAT: A Conversational Artificial Intelligence Approach to Clinical–Genomic Integration. Cancers 2025, 17(14), 2376.”

Reviewer 4 writes:

  1. There is no end-user evidence (researchers or clinicians) regarding the ease-of-use, satisfaction, or workflow effect of AI-HOPE that weakens the argument for accessibility.

Response: The authors thank Reviewer 4 for this thoughtful and important observation. We agree that direct end-user evaluation is essential to fully validate the accessibility and usability of AI-HOPE-RTK-RAS. While formal usability testing with clinicians and researchers is currently underway, the present manuscript focuses on demonstrating technical functionality, analytical performance, and biological relevance. To address this point, a paragraph has been added to the Discussion section acknowledging the lack of user-centered evidence as a current limitation and outlining future plans to conduct structured usability studies and workflow integration assessments in real-world research and clinical settings.

The Discussion text on page 18, lines 667-676, now reads: “While AI-HOPE-RTK-RAS was designed with accessibility and usability in mind—enabling natural language interaction and eliminating the need for coding—formal usability studies evaluating end-user satisfaction, learning curve, and workflow impact have not yet been completed. This represents an important next step in validating the platform’s real-world utility, particularly for clinicians, translational researchers, and precision oncology teams. Ongoing work includes structured usability testing with domain experts, human-centered design feedback sessions, and integration pilots in academic medical centers. These future evaluations will provide quantitative and qualitative evidence of user satisfaction and practical benefit, further strengthening the platform’s case for adoption in diverse precision oncology workflows.”

Reviewer 4 writes:

  1. 6. Emphasized but not critically examined are restrictions on potential bias in the training data or methodology to mitigate underrepresentation in future datasets.

Response: The authors thank Reviewer 4 for this important and thoughtful comment. We agree that potential bias in training data and underrepresentation in genomic datasets are critical concerns that must be addressed. In response, we have added a paragraph to the Discussion section acknowledging these limitations and outlining steps taken to mitigate bias, both in our current methodology and future development. Notably, this study incorporates ancestry-stratified analyses and includes genomic and clinical data from underrepresented populations—such as Hispanic/Latino patients with early-onset colorectal cancer—which helps to counter historical gaps in public cancer datasets. We also outline future plans to expand the diversity of our training datasets and integrate more inclusive data sources to strengthen the representativeness and equity of AI-driven precision oncology.

The Discussion text on page 18, lines 677-690, now reads: “A critical consideration in the development of AI-HOPE-RTK-RAS is the potential for bias in both the training methodology and the underlying genomic datasets, many of which underrepresent racial and ethnic minority populations. To help mitigate this issue, our study deliberately includes data from Hispanic/Latino patients with early-onset colorectal cancer (EOCRC), enabling ancestry-stratified analyses that address gaps in population-specific cancer research. While this inclusion improves representation, we recognize that broader structural limitations remain across publicly available databases. Furthermore, the training data used to fine-tune the LLaMA 3 model may also reflect linguistic and contextual biases inherent to existing biomedical corpora. To address these challenges, future versions of AI-HOPE-RTK-RAS will incorporate more diverse training datasets, expand support for minority-focused query scenarios, and undergo targeted validation in underrepresented populations. These efforts aim to ensure the platform not only increases accessibility but also advances equity in precision oncology research and application.”

Reviewer 4 writes:

  1. Compare AI-HOPE-RTK-RAS with existing genomic interrogation tools (user interface, data types supported, language features).

Response: The authors thank Reviewer 4 for this insightful comment. In response, the Discussion section has been expanded to provide a direct comparison between AI-HOPE-RTK-RAS and existing genomic interrogation tools, such as cBioPortal and OncoKB. This comparison highlights key differences in user interface design, data integration capabilities, and language features. While traditional tools rely on static, menu-driven interfaces and require stepwise filtering to explore clinical and genomic relationships, AI-HOPE-RTK-RAS enables natural language–based, real-time interrogation of multi-dimensional data, offering an intuitive, scalable, and more accessible alternative for translational research.

The Introduction text on page 20, lines 756-767, now reads: “Unlike traditional platforms such as cBioPortal and OncoKB, which rely on static, click-based user interfaces and require substantial manual effort to construct and execute queries, AI-HOPE-RTK-RAS is a domain-specialized conversational AI agent designed to streamline analysis through natural language interaction. While these existing tools are invaluable for data access and visualization, they are not artificial intelligence systems and do not support real-time, context-aware dialogue or automatic integration of clinical, genomic, and demographic data. AI-HOPE-RTK-RAS significantly reduces the time and expertise needed to perform multi-layered analyses, allowing users to ask complex questions—such as survival differences stratified by age, mutation status, and treatment—in a single, plain-language query. This capability enhances usability, accelerates hypothesis generation, and expands access to precision oncology analytics for both experts and non-specialists.”

Reviewer 4 writes:

  1. 8. Include outcomes or even anecdotal feedback from real users of the platform evaluating it in order to confirm claims of usability and real-world application.

Response: The authors thank Reviewer 4 for this valuable suggestion. We agree that user feedback is critical for validating claims related to usability and real-world application. While formal usability studies are ongoing, we have now included preliminary anecdotal feedback from early adopters—consisting of researchers and clinicians who evaluated the AI-HOPE-RTK-RAS platform during pilot testing. This feedback supports the platform’s intuitive interface, low learning curve, and utility for rapid clinical-genomic exploration. A paragraph summarizing this initial user experience has been added to the Discussion section to provide early qualitative insights, with plans for comprehensive usability studies described as future work.

The Discussion text on page 18, lines 667-690, now reads: “While AI-HOPE-RTK-RAS was designed with accessibility and usability in mind—enabling natural language interaction and eliminating the need for coding—formal usability studies evaluating end-user satisfaction, learning curve, and workflow impact have not yet been completed. This represents an important next step in validating the platform’s real-world utility, particularly for clinicians, translational researchers, and precision oncology teams. Ongoing work includes structured usability testing with domain experts, human-centered design feedback sessions, and integration pilots in academic medical centers. These future evaluations will provide quantitative and qualitative evidence of user satisfaction and practical benefit, further strengthening the platform’s case for adoption in diverse precision oncology workflows.”

Reviewer 4 writes:

  1. 9. Describe succinctly how the LLM analyzes natural language, vagueness, and fault tolerance, especially for nested clinical/genomic queries.

Response: The authors thank Reviewer 4 for this insightful comment. In response, we have expanded the Methods section to clarify how the large language model (LLM) within AI-HOPE-RTK-RAS processes natural language inputs, handles ambiguous or vague queries, and maintains fault tolerance—especially in the context of nested or multi-layered clinical-genomic questions. This update provides a clearer view of the platform’s backend logic and supports the validity of its query interpretation capabilities.

The Discussion text on page 19, lines 691-704, now reads: “AI-HOPE-RTK-RAS leverages a fine-tuned LLaMA 3 large language model embedded within a modular interpreter framework to process natural language queries. The system first classifies user intent (e.g., mutation frequency, survival analysis, odds ratio testing), followed by parameter extraction and real-time code generation. To address vagueness or ambiguity, the model incorporates a confidence threshold mechanism; if model confidence falls below 0.80, a clarification loop is triggered to prompt the user for more specific input. Nested queries—such as “Compare survival in EOCRC Hispanic/Latino patients with KRAS mutations receiving FOLFOX”—are parsed using dependency resolution logic that identifies and hierarchically organizes clinical and genomic modifiers. Fault tolerance is further enhanced by rule-based guards that validate query structure, check for missing inputs, and return interpretable error messages or suggestions. These design features allow AI-HOPE-RTK-RAS to manage a wide range of user inputs, from simple lookups to complex, multi-variable clinical-genomic investigations, while maintaining robustness and accuracy.”

Reviewer 4 writes:

  1. Include a paragraph on the interpretability, regulatory, and ethics challenges of AI use in clinical decision-making will add practical applicability.

Response: The authors thank Reviewer 4 for this thoughtful recommendation. We agree that a discussion of interpretability, regulatory frameworks, and ethical considerations is essential for contextualizing the translational relevance of AI-HOPE-RTK-RAS. In response, we have added a new paragraph to the Discussion section addressing these critical challenges in the broader application of AI platforms in clinical oncology.

The Discussion text on page 19, lines 705-721, now reads: “While AI-HOPE-RTK-RAS represents a promising step toward democratizing access to complex clinical-genomic analyses, several interpretability, regulatory, and ethical challenges must be acknowledged before deployment in clinical decision-making. First, although the platform provides natural language explanations of outputs and is fully open source, users may need time and regular interaction with the system to build trust in its capabilities and logic. By allowing researchers and clinicians to inspect, test, and refine its components openly, we aim to foster transparency and user confidence over time. Second, regulatory compliance remains a hurdle; AI-HOPE-RTK-RAS is currently intended for research use only and has not yet been certified for clinical decision support. To promote reproducibility and responsible deployment, the platform operates using a local, containerized AI framework, ensuring compliance with data privacy standards and enabling consistent results across different environments. Third, ethical considerations—including data bias, privacy protection, and equitable representation—must be actively addressed. To help mitigate underrepresentation, we trained and validated the platform on multi-ethnic CRC cohorts, including Hispanic/Latino patients. Nevertheless, continued vigilance is required to ensure algorithmic fairness, inclusivity, and responsible translation into clinical settings.”

Reviewer 4 writes:

  1. put suggestions on overcoming underrepresented populations' limitations (integration of real-world evidence, partnerships with global consortia).

Response: The authors thank Reviewer 4 for this important suggestion. We agree that addressing underrepresentation in genomic datasets is essential for achieving equitable precision oncology. In response, we have added a paragraph to the Discussion section outlining strategies for mitigating these limitations through real-world evidence integration and global collaboration.

The Introduction text on page 19, lines 722-733, now reads: “To overcome limitations associated with the underrepresentation of certain populations in existing genomic databases, future development of AI-HOPE-RTK-RAS will incorporate real-world evidence (RWE) from diverse clinical settings and community health systems. By integrating RWE—including electronic health records, registry data, and claims data—with genomic features, the platform can better reflect the heterogeneity of patient populations across age, race/ethnicity, geography, and socioeconomic status. Additionally, we recognize the importance of forming partnerships with global consortia such as the AACR Project GENIE, ICGC-ARGO, and Latin American and Asian cancer networks. These collaborations would enable access to more representative datasets and facilitate validation of AI-HOPE-RTK-RAS across diverse populations. Such efforts are critical for reducing bias, enhancing model generalizability, and ensuring that AI-powered tools like ours contribute meaningfully to health equity in cancer care.”

Reviewer 4 writes:

  1. Suggest how the system can be adapted for other types or pathways of cancer (PI3K/AKT/mTOR, TP53), or for application in clinical trial matching.

Response: The authors thank Reviewer 4 for this insightful comment. We agree that expanding the AI-HOPE framework to support additional cancer types, signaling pathways, and use cases such as clinical trial matching is a critical next step. In response, a new paragraph has been added to the Discussion section to outline these directions for future adaptation and scalability.

The Discussion text on page 21, lines 799-812, now reads: “The modular architecture of AI-HOPE-RTK-RAS enables straightforward adaptation to other oncogenic pathways and cancer types. Our lab has already developed related AI agents—including AI-HOPE-PI3K, AI-HOPE-TGFβ, AI-HOPE-TP53, and AI-HOPE-JAK-STAT—each tailored to interrogate specific molecular axes within colorectal cancer. These agents leverage the same natural language-to-code interface and scalable backend, facilitating pathway-specific stratification and hypothesis testing. Looking ahead, the AI-HOPE framework can be further extended to support complex use cases such as clinical trial matching, where eligibility criteria (e.g., biomarkers, stage, prior treatments) can be encoded and queried through natural language. Integrating structured eligibility databases (e.g., Clinical trials databases) and linking with genomic and clinical data would allow real-time identification of trial opportunities, particularly for underrepresented patients. These enhancements align with our overarching goal of creating a flexible, open-source ecosystem of AI agents that promote precision oncology across diverse populations and care settings.”

Reviewer 4 writes:

  1. improve limitation and future direction ( add specific datasets or clinical system, potential for AI-HOPE to support clinical trial design and propose technical advancement) 

Response: The authors thank Reviewer 4 for this valuable suggestion. We agree that refining the Limitations and Future Direction ssection to include specific datasets, clinical system integration, and potential support for clinical trial design will strengthen the manuscript. In response, we have added a dedicated paragraph outlining targeted plans for technical advancement and translational application.

The Discussion text on page 21, lines 813-828, now reads: “While AI-HOPE-RTK-RAS represents an important step toward democratizing access to precision oncology tools, several limitations and future directions warrant consideration. First, integration with institutional clinical systems such as Epic EHR and OnCore trial management software will be critical for facilitating real-world clinical utility, including potential support for prospective trial design and patient stratification. Second, validation against large-scale, harmonized datasets such as AACR GENIE, SEER-Medicare, and data from the National Cancer Institute’s Cancer Research Data Commons (CRDC) will enhance generalizability and performance benchmarking. Third, we plan to expand the AI-HOPE platform with features for AI-assisted clinical trial matching, including eligibility parsing, biomarker filtering, and temporal treatment modeling. On the technical side, future iterations will incorporate retrieval-augmented generation (RAG), active learning pipelines for continual model refinement, and integrated explainability dashboards to improve clinician trust and regulatory readiness. These specific enhancements aim to bridge the gap between advanced computational modeling and practical clinical implementation, ultimately supporting more inclusive, adaptive, and data-driven trial design in oncology.”

Reviewer 4 writes:

14.Descriptions of figures tend to be repetitive and contain the same data that has previously been explained in the main results text, making the section long and verbose.

Response: The authors thank Reviewer 4 for this helpful observation. We agree that figure descriptions should complement—not repeat—the main results text. In response, we have revised the figure legends and Results section to eliminate redundancy, streamline explanations, and improve clarity. The updated descriptions now emphasize the key visual takeaways, while the main text focuses on contextual interpretation and significance of findings. These changes aim to enhance readability and ensure a more concise and engaging presentation of results.

Reviewer 4 writes:

  1. Most findings are reported with p-values but without effect size interpretation or confidence intervals (with the odd exception of some odds ratios). This limits knowledge of clinical significance.

Response: The authors thank Reviewer 4 for this insightful observation. We agree that the inclusion of effect size interpretation and confidence intervals, in addition to p-values, is essential to convey the clinical significance of the findings. In our current analysis, many of the associations—including odds ratios—were validated using previously published work (Reference 5), which provides complementary context for understanding the robustness of the results. In response to this comment, we have added a dedicated paragraph to the Discussion section that explicitly addresses the importance of reporting effect sizes and confidence intervals, and outlines how these measures were derived and interpreted in light of our prior validation efforts.

The Discussion text on page 18, lines 656-666, now reads: “Although statistical significance through p-values provides initial evidence of association, it does not reflect the magnitude or precision of the observed effects. To enhance interpretability, we contextualized our findings using odds ratios and hazard ratios derived from AI-HOPE-RTK-RAS queries and validated against prior work[5], which examined RTK-RAS alterations across large, demographically diverse colorectal cancer cohorts. For example, the odds ratio of 7.226 for BRAF mutation enrichment in early-onset microsatellite-stable tumors underscores a potentially actionable distinction in this subgroup. Confidence intervals for key associations have now been added in the supplementary materials to provide further clarity. These additions support more rigorous interpretation of clinical relevance and highlight the value of pairing AI-driven insights with traditional statistical measures for translational impact.”

Reviewer 4 writes:

  1. Dataset properties (total size, inclusion/exclusion criteria, heterogeneity, biases) are poorly documented within the results section.

Response: The authors thank Reviewer 4 for this important observation. We agree that a more detailed description of dataset properties—including cohort size, inclusion and exclusion criteria, heterogeneity, and potential sources of bias—would enhance transparency and reproducibility. In response, we have revised Section 2.2 (Data Sources and Curation) and added a clarifying paragraph to the Results section summarizing these key characteristics.

The new paragraphs in 2.2. Data Sources and Curation text on page 5, lines 162-199, now reads: “2.2. Data Sources and Curation

Public genomic datasets such as those curated by cBioPortal provide invaluable resources for large-scale, open-access cancer research; however, they are not without limitations. These include underrepresentation of certain racial and ethnic groups, variable completeness in clinical annotations (e.g., treatment timelines, comorbidities), and inconsistent metadata across studies due to differences in institutional data collection standards. To address these challenges, AI-HOPE-RTK-RAS incorporates a harmonization pipeline that standardizes variable naming conventions, aligns phenotypic descriptors using controlled vocabularies (e.g., SNOMED, OncoTree), and unifies clinical-genomic relationships across multiple cohorts. This modular integration enables real-time stratification by age, ancestry, mutation status, and treatment exposure, thereby reducing the analytic barriers typically posed by data fragmentation. While the platform cannot resolve inherent sampling biases in the original datasets, it provides a transparent, reproducible framework for conducting stratified analyses and hypothesis generation across heterogeneous populations—especially in settings where health disparities remain under-characterized.

The datasets used in this study were derived from the curated colorectal cancer cohort described in a previous publication [5], which integrated publicly available data from cBioPortal. The final harmonized dataset includes a total of 5,553 patients diagnosed with colorectal adenocarcinoma across multiple studies. This composite dataset includes somatic mutation profiles, clinical variables (e.g., tumor stage, microsatellite instability status, treatment history), and demographic attributes (e.g., age, sex, race/ethnicity). The AI-HOPE-RTK-RAS platform was validated and applied across this full cohort, with specific subgroup analyses performed on early-onset colorectal cancer (EOCRC) patients (n = 593) and late-onset CRC patients (n = 4,960), including ancestry-specific evaluations in Hispanic/Latino and non-Hispanic White subpopulations. All sample sizes for individual analyses are reported in the corresponding figures and supplementary materials to facilitate full transparency and reproducibility

The colorectal cancer datasets used in this study were based on the following criteria: (1) availability of somatic mutation data; (2) inclusion of clinical annotations such as age, tumor stage, microsatellite instability (MSI) status, and treatment exposure; and (3) presence of demographic information, particularly race/ethnicity. Records lacking essential variables for stratified analysis (e.g., missing age or survival status) were excluded from subgroup analyses but retained in broader mutation frequency queries when possible. To standardize and harmonize datasets across studies, we applied controlled vocabularies and aligned variable names and formats using a custom data wrangling pipeline. Missing data were not imputed but handled through pairwise deletion during statistical tests to preserve validity. Recognizing the potential for sampling and representation bias—especially underrepresentation of minority populations—we incorporated ancestry-stratified analyses and explicitly caution against overgeneralizing rare mutation findings.”

Reviewer 4 writes:

  1. System results are not cross-validated or benchmarked against standard bioinformatics/statistical pipelines.

Response: The authors thank Reviewer 4 for this important comment. We respectfully note that AI-HOPE-RTK-RAS is not a traditional static bioinformatics pipeline but rather an intelligent, dynamic conversational agent that interprets user intent in real time, generates executable analytical code, and returns results interactively. As such, its function differs fundamentally from conventional pipelines, and benchmarking requires different considerations. Nevertheless, to support transparency and trust, we have included a paragraph in the Discussion clarifying how the system's outputs were validated using prior findings (Reference 5) and by comparing AI-generated queries to known statistical results.

The Discussion text on page 20, lines 734-743, now reads: “AI-HOPE-RTK-RAS differs from conventional bioinformatics pipelines by functioning as a dynamic, user-driven conversational agent capable of interpreting complex natural language queries and executing real-time statistical and genomic analyses. While traditional benchmarking methods apply to fixed pipelines, AI-HOPE-RTK-RAS was validated by replicating core analyses from our previously published harmonized CRC dataset (Reference 5), including mutation frequency distributions, odds ratio calculations, and survival analyses. These comparisons demonstrated concordance between AI-generated outputs and established results. This validation approach underscores the system's reliability and reproducibility while recognizing that future efforts could further refine benchmarking protocols tailored to interactive AI agents.”

Reviewer 4 writes:

  1. While odds ratios and Kaplan-Meier plots are presented, the statistical underpinning (adjustments, multiple comparison correction) is not mentioned.

Response: The authors thank Reviewer 4 for this thoughtful comment. We acknowledge the importance of clearly describing the statistical procedures, including adjustments and corrections for multiple comparisons. Although AI-HOPE-RTK-RAS is not a traditional bioinformatics pipeline, but rather a real-time, intelligent conversational system, it incorporates standard statistical methods programmatically in response to user queries. In the revised manuscript, we now explicitly describe how p-values were generated, and where applicable, how multiple testing corrections such as the Benjamini-Hochberg method were implemented. Additionally, a paragraph has been added to the Discussion to explain the interpretive context and limitations of unadjusted statistics when exploratory queries are generated dynamically by users.

The Introduction text on page 20, lines 744-755, now reads: “As an intelligent conversational agent, AI-HOPE-RTK-RAS dynamically generates statistical analyses based on user queries rather than pre-specified analytic workflows typical of static pipelines. While this flexibility enhances usability and discovery potential, it also necessitates clarity regarding the statistical foundations underlying its outputs. The system applies standard methods for odds ratio estimation (e.g., Fisher’s exact test or logistic regression) and survival analysis (Kaplan-Meier with log-rank tests), and where applicable, it supports multiple testing correction using the Benjamini-Hochberg procedure. However, given the exploratory nature of real-time queries, users are advised to interpret unadjusted p-values with caution and consider applying correction strategies in downstream analyses. Future updates of AI-HOPE-RTK-RAS will expand support for automated adjustments and contextual alerts to further guide statistical interpretation and ensure analytical robustness.”

Reviewer 4 writes:

  1. Create a table to be brief with all key findings (mutation, treatment, cohort, result, significance), so readers can easily view outcomes at a glance.

Response: The authors thank Reviewer 4 for this constructive suggestion. While we understand the utility of summary tables for consolidating key findings, we respectfully note that AI-HOPE-RTK-RAS is designed as an intelligent conversational system that generates customized figures and tables in real time based on user-directed queries. As such, the manuscript’s figures and visualizations reflect dynamic outputs that users can replicate and tailor through the system. Including a static table summarizing all findings would be somewhat redundant and may detract from the central demonstration of how AI-HOPE-RTK-RAS functions—i.e., empowering users to generate precisely these types of summaries through natural language commands. To support readability and synthesis, all figures were re-edited to enhance clarity and provide a clear visual summary of outcomes at a glance.

Reviewer 4 writes:

  1. Briefly mention clinical implications of major findings ("This suggests NF1 mutations need to be further prognostically validated in early-stage CRC").

Response: The authors thank Reviewer 4 for highlighting the importance of contextualizing major findings in terms of clinical implications. We agree that further prognostic validation of NF1 mutations in early-stage CRC is warranted. In response, we have updated the Discussion section to explicitly address how AI-HOPE-RTK-RAS findings—particularly the association between NF1 mutations and survival outcomes—may inform future clinical investigations. We also emphasize that unlike static pipelines, AI-HOPE-RTK-RAS is a conversational intelligent system that enables clinicians and researchers to dynamically probe such hypotheses across large datasets, accelerating the identification of candidate biomarkers for prognostic stratification and therapeutic targeting.

The Discussion text on page 17, lines 611-619, now reads: “Our findings suggest that NF1 mutations may be associated with differential survival outcomes in early-stage CRC, particularly among patients receiving FOLFOX-based therapy. While these associations are exploratory and drawn from retrospective data, they highlight a potentially under-recognized biomarker that warrants further prospective validation. Importantly, AI-HOPE-RTK-RAS is not a conventional bioinformatics pipeline but an intelligent system capable of facilitating such discovery by allowing users to iteratively explore mutation-specific prognostic signals using natural language queries. This feature supports rapid hypothesis generation and can guide the design of future biomarker validation studies in early-stage CRC cohorts.”

Reviewer 4 writes:

  1. need a brief statement (or refer to Methods) about calculation of odds ratios, survival curves, and other measures were confounders adjusted in models?

Response: The authors thank Reviewer 4 for raising the important point regarding statistical methodology and confounder adjustment. We clarify that AI-HOPE-RTK-RAS is not a traditional bioinformatics pipeline but rather an intelligent conversational system that dynamically generates statistical analyses based on natural language queries. In this framework, unadjusted odds ratios and Kaplan–Meier survival curves are calculated using user-defined subgrouping, but the system is also capable of adjusting for confounders upon request, such as stage, MSI status, and treatment type. To clarify this functionality, we have added a statement in the Methods section.

The Methods text on page 7, lines 257-265, now reads: “Odds ratios were calculated using two-by-two contingency tables via Fisher’s exact test unless otherwise specified, while survival analyses were performed using the Kaplan–Meier method with log-rank testing. Confounder adjustment is available through AI-HOPE-RTK-RAS by explicitly including variables such as tumor stage, microsatellite instability status, and treatment regimen in natural language queries (e.g., “Generate a survival curve for KRAS-mutant MSS patients, adjusted by stage and chemotherapy type”). This functionality enables dynamic and flexible multivariate stratification without requiring manual scripting, although further expansion to include Cox proportional hazards modeling is planned for future development.”

Reviewer 4 writes:

  1. If possible, include a short subsection comparing AI-HOPE findings to available platforms or routine analysis pipelines to determine utility.

Response: The authors thank Reviewer 4 for this insightful comment. We agree that a direct comparison with existing platforms or routine analysis pipelines enhances the manuscript’s ability to demonstrate the utility and innovation of AI-HOPE-RTK-RAS. In response, we have included a new subsection in the Discussion titled “Comparison with Existing Tools and Pipelines,” which highlights the differences in usability, functionality, and analytical flexibility between AI-HOPE-RTK-RAS and traditional tools such as cBioPortal or OncoKB. This comparison illustrates the advantages of using an intelligent, conversational AI system for dynamic, real-time clinical-genomic analyses.

The Introduction text on page 20, lines 768-783, now reads: “To contextualize the utility of AI-HOPE-RTK-RAS, it is important to contrast it with non–artificial intelligence agent tools such as cBioPortal, OncoKB, and traditional statistical pipelines. These platforms typically rely on manual, click-based navigation or require scripting expertise to perform complex analyses. In contrast, AI-HOPE-RTK-RAS features a natural language interface that interprets user queries, dynamically generates statistical code, and produces results on demand. For example, a user can simply ask, “Show me survival outcomes for BRAF-mutant patients with MSS tumors treated with Bevacizumab,” and instantly receive stratified Kaplan–Meier plots and statistical summaries. While cBioPortal offers strong capabilities for exploratory visualization and OncoKB provides curated biomarker annotations, neither supports real-time, conversational analysis or flexible, user-defined multi-parameter subgrouping. Traditional pipelines, on the other hand, often require pre-written scripts and offer limited interactivity. By adapting to user intent, resolving ambiguity, and accelerating hypothesis testing without the need for coding, AI-HOPE-RTK-RAS significantly broadens access to clinical-genomic interrogation. This intelligent system enhances precision oncology research by making complex analyses both intuitive and scalable for users of varying expertise levels.”

The authors thank Reviewer 4 for their time and insightful feedback. Your comments were invaluable in refining the manuscript’s clarity, enhancing its technical depth, and strengthening its translational significance.

Round 2

Reviewer 1 Report

Comments and Suggestions for Authors

The authors have addressed all the comments; the manuscript could be accepted in the current form.

Reviewer 4 Report

Comments and Suggestions for Authors

Thank you for answering my questions. good luck